

**Impacts of absorbing aerosol deposition on snowpack and hydrologic**
**cycle in the Rocky Mountain region based on variable-resolution**
**CESM (VR-CESM) simulations**
Chenglai Wu[1,2], Xiaohong Liu[1,*], Zhaohui Lin[2,3], Stefan R. Rahimi-Esfarjani[1], and
Zheng Lu[1]
[1]*Department of Atmospheric Science, University of Wyoming, Laramie, Wyoming,*
*USA*
[2]*International Center for Climate and Environment Sciences, Institute of Atmospheric*
*Physics, Chinese Academy of Sciences, Beijing, China*
[3]*University of Chinese Academy of Sciences, Beijing, China*
*\*Corresponding to:*
Xiaohong Liu
Department of Atmospheric Science
University of Wyoming
Dept. 3038, 1000 East University Avenue
Laramie, WY 82071
Email: xliu6@uwyo.edu.





**Abstract**
Deposition of light-absorbing aerosols (LAAs) such as black carbon (BC) and dust
onto snow cover has been suggested to reduce the snow albedo, and modulate the
snowpack and consequent hydrologic cycle. In this study we use the
variable-resolution Community Earth System Model (VR-CESM) with a regionally
refined high-resolution (0.125º) grid to quantify the impacts of LAAs in snow in the
Rocky Mountain region during the period of 1981-2005. We first evaluate the model
simulation of LAA concentrations both near the surface and in snow, and then
investigate the snowpack and runoff changes induced by LAAs in snow. The model
simulates similar magnitudes of near-surface atmospheric dust concentrations as
observations. Although the model underestimates near-surface atmospheric BC
concentrations, the simulated BC-in-snow concentrations are overall comparable to
observations. Regional mean surface radiative effect (SRE) due to LAAs in snow
reaches up to 0.6-1.7 W m$^{-2}$ in spring, and dust contributes to about 21-43% of total
SRE. Due to positive snow-albedo feedbacks induced by the LAAs' SRE, snow water
equivalent reduces by 2-50 mm and snow cover fraction by 5-20% in the two regions
around the mountains (Eastern Snake River Plain and Southwestern Wyoming),
corresponding to an increase of surface air temperature by 0.9-1.1$^{\circ}$C. During the snow
melting period, LAAs accelerate the hydrologic cycle with monthly runoff increases
of 0.15-1.00 mm day$^{-1}$ in April-May and reductions of 0.04-0.18 mm day$^{-1}$ in
June-July in the mountainous regions. Of all the mountainous regions, Southern



Rockies experience the largest reduction of total runoff by 15% during the later stage
of snow melt (i.e., June and July). Our results highlight the potentially important role
of LAA interactions with snowpack and subsequent impacts on the hydrologic cycles
across the Rocky Mountains.
**1.  Introduction**

Water resources are essential to human society and economic development as

well as ecosystems in the western United States. Most of primary water resources in
the inland western U.S. come from the Rocky Mountains' snowpack (Serreze et al.,
1999). Therefore, to develop the water resource management strategy, it is necessary
to know the information of snow accumulation and snowmelt timing. Climate change
is an important factor influencing the snowpack in the Rocky Mountain region, as has
been shown in many previous studies (e.g., Abatzoglou, 2011; Pederson et al., 2011;
Rhoades et al., 2017). Another important factor is the light-absorbing aerosols (LAAs,
e.g., black carbon (BC), organic carbon (OC), and dust) in snow (e.g., Flanner et al.,
2007; Painter et al., 2007; Qian et al., 2015; Yasunari et al., 2015). Previous studies
have shown that LAAs in snow can significantly reduce the surface albedo (often
known as snow darkening effect (SDE)), modify the surface energy budget and
snowmelt, and lead to the modification of hydrologic cycles (e.g., Warren and
Wiscombe, 1980; Hansen and Nazarenko, 2004; Flanner et al., 2007, 2009; Painter et
al., 2007, 2010; Qian et al., 2009, 2011; Yasunari et al., 2015). Moreover, the



LAAs-induced snow albedo reduction may initiate positive feedback processes, which
can amplify the reduction of snowpack (e.g., Flanner et al., 2009; Qian et al., 2009).

In past decades modeling studies have been undertaken to quantify the impacts

of SDE by LAAs (e.g., Flanner et al., 2007; Qian et al., 2009; Oaida et al., 2015;
Yasunari et al., 2015). Generally the models they developed have the ability to
simulate the temporal evolution of snow albedo under the influence of LAAs in snow.
These studies have enhanced our understanding of the spatial and temporal variations
of climate forcings and responses due to LAAs in snow from regional scales (e.g.,
Qian et al., 2009; Oaida et al., 2015) to global scales (e.g., Flanner et al., 2007;
Yasunari et al., 2015). For example, the impacts of LAAs in snow are stronger in
regions with considerable snow cover and sufficient LAAs deposition (e.g., Arctic,
Northeast China, Tibetan Plateau, and western U.S.) than in other regions, and they
are largest during the snowmelt period due to the positive snow-albedo feedback.
However, as also mentioned in these studies, reliable quantification of impacts of
LAAs in snow is hindered by the model deficiencies in simulating the snowpack and
aerosol cycles, with additional uncertainties induced by the parameterization of
snow-aerosol-radiation interactions.

In particular, previous studies used the coarse-resolution global climate

models (GCMs) or high-resolution regional climate models (RCMs) to quantify the
impacts of LAAs in snow. However, there are weaknesses in either coarse-resolution
GCMs or RCMs. Both snowfall and snow accumulation depend on the temperature



and precipitation, and thus distribution of snowpack depends strongly on topographic
variablility. Current GCMs with a typical horizontal resolution of 1° to 2° cannot
resolve the snowpack over the regions with complex terrains (e.g., Rocky Mountains)
due to the coarse resolution (Rhoades et al., 2016; Wu et al., 2017), which impedes
the reliable quantifications of SDE by LAAs in mountainous regions (e.g., Flanner et
al., 2007; Yasunari et al., 2015). For RCMs, they can simulate the snowpack more
accurately but are not able to simulate the global transport of aerosols to the focused
region except that aerosol transport along the boundary is prescribed (e.g., Qian et al.,
2009; Oaida et al., 2015). Moreover, LAAs in snow may also influence the climate
beyond the focused region (e.g., Yasunari et al., 2015), which cannot be accounted for
in RCMs. Variable resolution GCMs (VR-GCMs) can overcome these weaknesses of
either coarse-resolution GCMs or RCMs and serve as a better tool to quantify the
impacts of LAAs in snow. Although GCMs with globally uniform high resolutions
(10-30 km) may be an ideal tool to simulate the snowpack and snow-aerosol-radiation
interactions, they are not widely applied due to the constraints of computational
resources (e.g., Haarsma et al., 2016). Instead, using VR-GCMs is a more economic
approach and has gained increasing utility in recent years (e.g., Zarzycki et al, 2014a,
b; Sakaguchi et al., 2015).

A variable-resolution version of the Community Earth System Model

(VR-CESM) has been developed (Zarzycki et al., 2014a, b). With a refined high
resolution, VR-CESM has shown significant improvements of the Atlantic tropical



storms (Zarzycki and Jablonowski, 2014) and South America orographic precipitation
(Zarzycki et al., 2015). The model has also been used in the regional climate
simulations in western U.S., and results show that VR-CESM is capable of
reproducing the spatial patterns and the seasonal evolution of temperature,
precipitation, and snowpack in Sierra Nevada (Huang et al., 2016; Rhoades et al.,
2016) and in Rocky Mountains (Wu et al., 2017). In particular, VR-CESM simulates
reasonably the magnitude of snow water equivalent, the timing of snow water
equivalent peaks, and the duration of snow cover in the Rocky Mountains by
comparing against the Snow Telemetry (SNOTEL) and MODIS (Moderate
Resolution Imaging Spectroradiometer) snow cover observations (Wu et al., 2017).

Following the evaluation study of Wu et al. (2017), here we use VR-CESM to

investigate the impacts of LAAs in snow (BC and dust) on the snowpack and
hydrologic cycles over the Rocky Mountains. By comparing the two VR-CESM
simulations with and without considering the impacts of LAAs in snow, we examine
the changes in surface radiative transfer, temperature, snowpack, and runoff induced
by LAAs in snow. To our knowledge, it is the first time that VR-CESM is applied for
the study of LAAs in snow. Our results will demonstrate that VR-CESM is skillful for
this kind of research.

The remainder of the paper is organized as follows. Section 2 introduces the

model and experimental design. Section 3 describes the observation data used for
validation of model simulations of aerosol fields in the surface air and in snow.



Section 4 presents the evaluation of aerosols fields, followed by their surface radiative
effect (SRE), as well as the change of surface temperature, snowpack, and runoff
induced by LAAs in snow. Discussion and conclusions are given in section 5.
**2.   Model and experimental design**

The model used in this study is VR-CESM, a version of CESM (version 1.2.0)

with the variable-resolution capability (Zarzycki et al., 2014a, b). CESM is a
state-of-the-art Earth system modeling framework that allows for for investigations of
a diverse set of Earth system interactions across multiple time and space scales
(Hurrell et al., 2013). CESM uses the Community Atmosphere Model version 5
(CAM5) for the atmospheric component (Neale et al., 2010). The variable-resolution
capability is implemented into the Spectral Element (SE) dynamic core of CAM5.
The SE dynamic core uses a continuous Galerkin spectral finite-element method
designed for fully unstructured quadrilateral meshes, and has demonstrated
near-optimal (close to linear) parallel scalability on tens of thousands of cores (Dennis
et al., 2012). This enables the model to run efficiently on decadal to multi-decadal
time scales. For the land component, CESM uses the Community Land Model version
4 (CLM4). CLM4 can be run at the same horizontal resolutions as CAM5 and thus
also benefit from the variable-resolution capability of CAM5.

CESM also includes advanced physics for CAM5 (Neale et al., 2010) and

CLM4 (Oleson et al., 2010). The CAM5 physics suite consists of shallow convection
(Park and Bretherton, 2009), deep convection (Zhang and McFarlane, 1995; Richter



and Rasch, 2008), cloud microphysics (Morrison and Gettelman 2008) and
macrophysics (Park et al. 2014), radiation (Iacono et al. 2008), and aerosols (Liu et al.,
2012). For aerosols, a modal aerosol module (MAM) is adopted to represent the
internal and external mixing of aerosol components such as BC, OC, sulfate,
ammonium, sea salt, and mineral dust (Liu et al., 2012). CLM4 physics includes a
suite of parameterizations for land-atmosphere exchange of water, energy and
chemical compounds. In particular, CLM4 explicitly represents the snowpack (snow
accumulation and melt) by a snow model and its coupling with the SNow, Ice and
Aerosol Radiation (SNICAR) model for snow-aerosol-climate interactions (Flanner et
al., 2007). SNICAR incorporates a two-stream radiative transfer solution of Toon et
al. (1989) to calculate the snow albedo and the vertical absorption profile from solar
zenith angle, albedo of the substrate underlying snow, mass concentrations of
atmospheric-deposited aerosols (BC and dust), and ice effective grain size ($r_e$). $r_e$ is
simulated with a snow aging routine (Oleson et al., 2010). SNICAR is compatible
with the new modal aerosol module of CAM5 (Liu et al., 2012) in the treatment of
aerosol deposition (Flanner et al., 2012). As our knowledge of OC optical properties
is limited, the impact of absorbing OC on snow albedo is not included in the standard
CLM4 and thus not considered in this study.

For the high-resolution modeling, we have designed a variable-resolution grid

that transits from global quasi-uniform 1º resolution to a refined 0.125º resolution in
the Rocky Mountains (Figure 1a). The variable-resolution grid is the same as that



used in Wu et al. (2017), and is generated by the open-source software package called
SQuadGen (Ullrich, 2014). A topographical dataset for this variable-resolution grid is
also generated accordingly by the National Center for Atmospheric Research (NCAR)
global model topography generation software called NCAR_Topo (v1.0) (Lauritzen et
al., 2015) as described in Wu et al. (2017). As shown in Figure 1b, the high-resolution
grids resolve well the variations of terrain in the Rocky Mountains.    Note that the
standard CESM using the coarse 1º grids cannot resolve the fine-scale variations of
terrain in the Rocky Mountains (see Figure 2 of Wu et al. (2017)). In Wu et al. (2017),
we have shown that VR-CESM performs well in the simulation of regional climate
patterns, including spatial distributions and seasonal evolution of temperature,
precipitation, and snowpack in the Rocky Mountain region. In this study, we further
apply VR-CESM to simulate the SDE of LAAs and its impacts on snowpack and
hydrologic cycles in the Rocky Mountains.

VR-CESM is run in the coupled land-atmosphere mode with prescribed

observed monthly 1º×1º sea surface temperature and sea ice coverage (Hurrell et al.,
2008), following the Atmospheric Model Intercomparison Project (AMIP) protocols
(Gates, 1992). The simulation period is from 1979 to 2005, and the results for the last
25 years (1981-2005) are used for the analysis shown below. Historical greenhouse
gas concentrations, and anthropogenic aerosol and precursor gas emissions are
prescribed from the datasets of Lamarque et al. (2010). In particular, the BC
emissions consist of various sources, including domestic, energy, transportation,





waste, shipping, and wildfire (forest and grass fires) emissions. We note that the
horizontal resolution for BC emission used in this study is 1.9×2.5º. The relatively
coarse resolution of BC emission may partly explain the model's bias in the
simulation of BC concentrations near the surface and in snow across regions where
local BC sources can contribute significantly to the observed BC concentrations, as
will be discussed in section 4.

For dust aerosol, the emission flux is calculated interactively in the model at

each time step by a dust emission scheme (Oleson et al., 2010). The dust emission
flux is calculated from the friction velocity, threshold friction velocity, atmospheric
density, clay content in the soil, areal fraction of exposed bare soil, and source
erodibility (Oleson et al., 2010; Wu et al., 2016). Due to the large uncertainty in
modeled dust emission, the dust emission scheme also adopts a posterior tuning factor
($T$) to simulate the reasonable dust emission amount. With the increase of model
resolution, VR-CESM produces much higher dust concentrations compared to the
observations (section 3) in North America if $T$ used in the standard CESM with
quasi-uniform 1º resolution is used. Therefore, for VR-CESM, $T$ is reduced by a
factor of 2.6 to produce the similar magnitudes of near-surface dust concentrations as
the observations, as will be shown in section 4.1. Note that such a reduction of $T$ is
only applied in North America, since other continents have a resolution of
quasi-uniform 1º, the same as in the standard CESM.



In addition to the control experiment with the impacts of LAAs (BC and dust)
in snow included (CTL), we conduct a sensitivity experiment that turns off the impact
of LAAs in snow (NoSDE). Through the comparison of these two simulations (CTL
and NoSDE), the impacts of SDE by LAAs on the snowpack and hydrologic cycles
can be identified. To facilitate the analysis of SDE, we also calculate the surface
radiative effect (SRE) by BC (dust) in snow in the control experiment from the
difference of absorbed radiation with all aerosols (i.e., the standard radiation call) and
with all aerosols except BC (dust) as Flanner et al. (2007) (a diagnostic radiation call).
To quantify the impacts of LAAs in snow, we mainly focus on the five regions.
Three of these regions are in the high mountains: Northern Rockies, Greater
Yellowstone region, and Southern Rockies. The elevation is higher in Greater
Yellowstone region and Southern Rockies (>2250 m) than in Northern Rockies (>750
m). The other two regions are over the plains near the mountains: Snake River Basin
and Southwestern Wyoming. These two regions are selected because they are close to
the source regions of BC and dust and also have considerable snow cover (>50%) in
winter. These five regions are shown in Figure 1c.
**3.  Observations**
We will use various observations to validate the model simulation of aerosol
(BC and dust) concentrations near the surface and in snow.
First, we use the observations of near-surface BC and dust concentrations from
the Interagency Monitoring of PROtected Visual Environments (IMPROVE) network



(Malm et al., 1994). Observed mass concentrations of Elemental Carbon (EC) are
used for the comparison with model simulation of BC concentrations. Although EC
can be somewhat different from BC (Andreae and Gelencser, 2006), EC
concentrations have been widely used for the validation of BC concentrations in
previous studies (e.g., Koch et al., 2009; Liu et al., 2012). For dust, simulated dust
concentration accounts for dust particles with diameters below 10 μm. To compare,
observed mass concentrations of fine soil (FS, with the diameter <2.5 μm) and coarse
mass (CM, with the diameter between 2.5 μm and 10 μm) from IMPROVE are
combined, following the approach of Kavouras et al. (2007) and Wells et al. (2007).
In reality, in addition to dust, CM may also contain other aerosols such as sulfate,
nitrate, organic and elemental carbon, and sea salt. However, according to the study of
Malm et al. (2007), who analyzed the speciation of coarse particles collected at nine
selected rural IMPROVE stations in 2004, the contributions of dust to CM are above
70% (74-90%) in the three stations in inland western U.S. In their study, lower
contributions of dust to CM (34% and 65%) were found in the two stations near the
coast. We caution that these two stations were <150 km away from the metropolitan
regions indicating that urban emissions may also contribute to CM there. Additional
contributions may result from sea salt or sodium nitrate resulting from reactions of
nitric acid with sea salt, as mentioned in their study (Malm et al., 2007). Therefore, to
minimize the contributions of other aerosols to CM, we do not use the stations in or
near the metropolitan regions or near the coast for the validation of dust concentration.



Nonetheless, we acknowledge that there may be small contributions from other
aerosols to CM and the estimated dust concentration by summing FS and CM may
represent an upper limit of dust concentrations (with the diameter <10 μm) from the
observations. Note that the observation period of IMPROVE varies with the stations,
some stations started earlier in 1980s, and some from more recently (2000s). To
derive a climatological dataset for model comparisons, we only select the stations
with more than 5 years of dust observations. Totally there are 80 and 94 stations for
BC and dust observations, respectively, in the western U.S. (Figure 2).

Second, we collect the field measurements of BC mass concentrations in snow

($C_{BC}$) from previously published studies. Although field observations of $C_{BC}$ in snow
extended back to 1980s, they were made mostly in high-latitudes, Alps Mountains,
Cascade Mountains, eastern Canada, and West Texas/New Mexico (see Qian et al.
(2015) and references therein). Recently, Doherty et al. (2014) made valuable
measurements of the vertical profiles of LAAs in seasonal snow from January to
March of 2013. They used an Integrating Sphere integrating SandWich (ISSW)
Spectrophotometer to estimate the BC mixing ratios in snow over 67 sites in North
America (including 17 sites in the Rocky Mountain region). Observed $C_{BC}$ by
Doherty et al. (2014) was recorded on a single day. Doherty et al. (2016) further
provided the temporal variations of $C_{BC}$ at four stations, three at Idaho (January to
March of 2014) and one at Utah (February to March of 2013 and 2014). Doherty et al.
(2016) also calibrated the ISSW measurements using an incandescence technique (the





Single Particle Soot Photometer, SP2) in a subset of the observations, which was
supposed to capture $C_{BC}$ more accurately, and derived a ratio of $C_{BC}$ by ISSW to $C_{BC}$
by SP2 based on their linear relationship for the estimation of real $C_{BC}$. This
calibration is applied to the dataset of Doherty et al. (2014) in our study, and thus the
observations of Doherty et al. (2014) and Doherty et al. (2016) are comparable and
used here.

In addition, Skiles and Painter (2016b) made daily measurements of

BC-in-snow with SP2 in the Senator Beck Basin Study Area (SBBSA) in the San
Juan Mountains during a period of two month (late March to middle May) in 2013.
The locations and sample dates as well as the measurements for these stations are
given in Table 1. If measurements of $C_{BC}$ on multiple days were made, the means and
standard deviations of $C_{BC}$ are given. As our simulation period does not encompass
the years 2013 and 2014, we will compare the monthly mean results for 25 years
(1981-2005) by focusing more on the general magnitudes and spatial distributions of
$C_{BC}$. At each station, the mean results for the month (or months) when the
observations were made, as well as the maximum and minimum $C_{BC}$, are derived
from the 25-year simulation and compared to the observations.

Third, for dust, there are few observations of dust mass mixing ratio in snow

($C_{dust}$) in the Rocky Mountain region. To our best knowledge, the only published
observations were conducted in the Senator Beck Basin Study Area (SBBSA) in the
San Juan Mountains (in Southern Rockies) with at least 9-year (2005-2013) records



(Painter et al., 2007, 2012; Skiles and Painter, 2016a, 2016b). Snow samples at a
depth of 30 cm were collected at irregular time intervals for each dust event from
March to May. We will compare simulated $C_{dust}$ for the whole snow columns with the
observations (for 0-30 cm depth), but acknowledge that $C_{dust}$ may vary in the snow
underneath 0-30 cm snow layer. Another consideration is that observed $C_{dust}$ contains
all the dust particles while simulated $C_{dust}$ only accounts for the dust particles with
diameters smaller than 10 μm. According to the observations of Reynolds et al.
(2016), the concentration of total suspended particles (TSP) in the atmosphere is
mainly contributed from particles with diameters larger than 10 μm. This will affect
the model comparison with the observations, which will be discussed in section 4.
**4.   Results**
**4.1 Spatial patterns of near-surface aerosol concentrations**

Before we examine the impacts of aerosol deposition onto snow, we will first

evaluate the aerosol simulations by the model. Figure 2 shows the spatial patterns of
cold season (winter and spring) mean emission fluxes and near-surface concentrations
of BC and dust in the western U.S. from the VR-CESM simulation. The IMPROVE
stations are also denoted by cycles with larger cycle sizes indicating higher observed
near-surface BC/dust concentrations. In the model, the BC emission flux is prescribed
and is largest in the Pacific Coast and southern Arizona. BC emission fluxes are
relatively large in central–northern Colorado and Northwestern Utah, where large
metropolises are located. Corresponding to the patterns of BC emission flux,


simulated near-surface BC concentrations (>100 ng m$^{-3}$) are also higher in these
regions. A band with relatively high near-surface BC concentrations around 50-100
ng m$^{-3}$ is also found in southern Idaho, to the west of the Greater Yellowstone region
and to the south of Northern Rockies, indicating the transportation of BC around the
mountains. Near-surface BC concentrations decrease at higher elevations. The spatial
patterns simulated by the model are generally consistent with observations, e.g.,
higher BC concentrations in the source regions and lower in the mountains.

Dust sources are located in the dry regions with exposed bare soils, such as the

southwestern U.S. (southern California, western Arizona, and southern New Mexico),
the northern Mexico, the Great Basin, and the Colorado Plateau. Dust emissions are
also found in the Great Plains, although it is much weaker. In the Great Plains
agricultural activities can disturb the soil, making it vulnerable to wind erosion
(Ginoux et al., 2012). Simulated cold season mean dust concentrations are higher
(10-500 μg m$^{-3}$) in the source regions, but decrease dramatically (0.1-5 μg m$^{-3}$) to the
mountains. Compared to the observations, the model reproduces the spatial patterns of
near-surface dust concentrations with higher concentrations in the southwestern part
of US. However, the model tends to overestimate the dust concentrations in Utah,
indicating that dust emission may be overestimated there.

Comparisons of modeled and observed near-surface BC/dust concentrations at

the IMPROVE stations are further shown in Figure 3. Overall, the model captures the
magnitudes of observed near-surface BC and dust concentrations with the differences



between observations and simulations within a factor of 5 for most of the stations.
The model simulates similar magnitudes of observed near-surface BC concentrations
at the stations along the West Coast and in the Southwestern U.S. However, the model
tends to underestimate observed near-surface BC concentrations in Utah–Nevada
regions, the Rocky Mountains, and the Great Plains, where the stations are located in
the downwind of source regions (Figure 2b). In particular, observed near-surface BC
concentrations are underestimated by about a factor of two in the Rocky Mountains.
The underestimation of near-surface BC concentrations in these regions may suggest
that transport of BC in our simulations is too weak. This deficiency may also be
ascribed to the local BC sources (e.g., Doherty et al., 2014) not resolved by the
prescribed BC emission in the model (e.g., at 1.9×2.5º resolution). For dust, although
the model overestimates near-surface dust concentrations for most of the stations near
the dust sources (southwestern U.S., Utah, and Nevada), the model simulates well the
magnitude of near-surface dust concentrations in the Rocky Mountains. This may also
be associated with underestimated transport in the model, as indicated in the bias of
near-surface BC concentrations in the downwind regions.

Note that although only the BC and dust emission fluxes over the western U.S.

are shown in Figure 2, long-range transport of these aerosols from other regions (e.g.,
Asia and Africa) can also contribute to BC (e.g., Zhang et al., 2015) and dust (Wells
et al., 2007) concentrations in the western U.S. Meanwhile, there are also substantial
variations of aerosol emission in the western U.S. As mentioned in section 2, although



we adopt VR-CESM with a refined high resolution (0.125°) in the Rocky Mountains,
we use a coarse resolution gridded emission dataset (i.e., 1.9°×2.5°) for BC. For dust,
the small-scale variations of dust emission can be represented in the model as it is
calculated online in the model. However, dust emission depends on many variables
such as near-surface winds, soil moisture, vegetation cover, and soil texture (Oleson
et al., 2010; Wu et al., 2016), which may be biased. In particular, in Utah and Nevada,
simulated near-surface dust concentrations are about 2-3 times as large as
observations, indicating the significant overestimation of dust emission in the region.
Despite the aforementioned biases, the model does reasonably well in the simulation
of spatial variations of near-surface BC and dust concentrations in western U.S.
**4.2 Aerosol-in-snow concentrations**
Figure 4 shows the spatial distributions of BC and dust mass mixing ratios in
snow in winter and spring from VR-CESM simulations. BC-in-snow mass mixing
ratio in the Rocky Mountains ranges from 2-50 ng g$^{-1}$, which is consistent with a
previous study (Qian et al., 2009). The dust-in-snow mass mixing ratio (0.1-50 µg g$^{-1}$)
is about 2-3 orders of magnitude higher than that of BC-in-snow. The spatial pattern
of BC-in-snow mixing ratios is consistent with that of near-surface atmospheric BC
concentration, which features higher values in northern Utah and southern Idaho and
lower values in the higher mountains (Figure 2b). Dust-in-snow mixing ratios are
higher in Utah and downwind regions (western Colorado and southern Idaho), which
is consistent with the distribution of near-surface atmospheric dust concentrations.



Dust-in-snow mixing ratio is also higher in the northern Great Plains, where dust
emission is also evident (Figure 2c). In addition, BC and dust mixing ratios are larger
(10-100 ng g$^{-1}$ and 2-50 μg g$^{-1}$, respectively) in the Southern Rockies than in Northern
Rockies and Greater Yellowstone region. BC and dust mass mixing ratios are smaller
in Greater Yellowstone region with ranges from 10-50 ng g$^{-1}$ and from 0.2-2 μg g$^{-1}$,
respectively, and are smallest in the Northern Rockies with the values below 20 ng g$^{-1}$
and below 2 μg g$^{-1}$, respectively. BC and dust mixing ratios in snow are larger in
spring than in winter in most of the Rocky Mountain region. This indicates that BC
and dust accumulate within the snow column during the snow accumulation and the
early portion of the snow melting season.

The comparison of BC mass mixing ratios in the snow column at the 17 sites

from VR-CESM simulations and observations is shown in Figure 5. As observations
only sampled the snow in one day or tens of days, simulated mean and standard
deviations of BC-in-snow mass mixing ratios in the same month are given for the
comparison. The months of observations occurred between January and May.
Observed BC-in-snow mass mixing ratios range from 5.6 ng g$^{-1}$ to 33.6 ng g$^{-1}$ at the
17 sites. The model reproduces reasonably the magnitude of observed BC-in-snow
mass mixing ratios at most of the stations. The exception is that at the two stations
(sites #15 and #16) where the model significantly underestimates observed
BC-in-snow mass mixing ratios by a factor of more than 5. Site #15 is close to the
sites #13 and #14 and they are all located on the southwestern flanks of Northern



Rockiest (Figure 4). Observed BC mass mixing ratios at site #15 (14.9 ng g$^{-1}$) is
higher than those at sites #14 (13.3 ng g$^{-1}$) and #13 (9.8 ng/g), which indicates the
northward transport of BC (Doherty et al., 2016). Simulated BC-in-snow mass mixing
ratios are around 8.8 ng g$^{-1}$ and 10.9 ng g$^{-1}$ at sites #13 and #14, respectively, which
are comparable to the observations. However, the simulated value is only 2.5 ng g$^{-1}$ at
site #15. This indicates that the model may lack the ability to simulate the transport
and/or deposition of BC in this region. Site 16 is located in northeastern Utah, where
snow depths are smaller compared to other mountains regions. When snow is melted
completely, BC-in-snow mixing ratio will be zero, but the model will average the
simulation results at different time steps to derive the mean result. In the observations,
only BC-in-snow samples are accounted for. The different sample strategy may partly
explain the difference between the simulations and observations. Another reason may
be that observations show a large interannual variability of BC-in-snow mass mixing
ratios at site #16 and the observation period was not long enough for the derivation of
a climatological mean as in the simulation.

Note that although near-surface BC concentrations in the atmosphere are

underestimated in the Rocky Mountains in the model (section 4.1), BC mass mixing
ratios in the snow are not overall underestimated. In our study, we calibrate the
observational data of Doherty et al. (2014) by using the correction factor based on the
comparison of ISSW and SP2, assuming that SP2 can more accurately measure the
mass mixing ratio of BC compared to ISSW (Doherty et al., 2016). However,





although SP2 can provide a direct measurement of BC, SP2 may underestimate the
real amount of BC-in-snow mass when BC is attached to larger particles (e.g., dust
and sea salt) or aggregates to large sizes in snow due to the size range (e.g., ~0.08-0.7
μm) limitations in SP2 (Qian et al., 2015). Because of this, the real amount of
BC-in-snow mass may be higher than that measured by SP2. Therefore, VR-CESM
may also underestimate the BC-in-snow mass mixing ratios, although the exact degree
of underestimation is unknown. Another reason for the inconsistency of BC mass
mixing ratios in snow and near-surface BC concentrations in the atmosphere may be
related to the snow aging/melting and BC-in-snow accumulation and flushing-out,
which are associated with large uncertainties (Flanner et al., 2007; Qian et al., 2014).

For dust in snow, the simulated mean dust mass mixing ratio in snow in spring

is 19.6 μg g$^{-1}$ in the San Juan Mountains. The simulated standard deviation, minimum,
and maximum of dust mass mixing ratios for 1981-2005 are 22.4 μg g$^{-1}$, 5.3 μg g$^{-1}$,
and 118.7 μg g$^{-1}$, respectively. This value is one to two orders of magnitude smaller
compared to the observations of Skiles and Painter et al. (2016a), which showed that,
at the end of the snow season, the total dust-in-snow mass mixing ratios range from
0.2 to 4.8 mg g$^{-1}$. Much smaller dust-in-snow mass mixing ratios in the simulations
may be ascribed to the fact that the model only accounts for dust particles with
diameters smaller than 10 μm, while the observations include all the sizes of dust
particles in the snow. Actually, an observation made by Reynolds et al. (2016) in the
Utah-Colorado region showed that concentrations of total suspended particles (TSP)





in the atmosphere are mainly contributed from larger particles with diameters larger
than 10 µm. Therefore, the model may underestimate the impacts of dust deposition
into snow, and the dust impacts in this study, which will be discussed below, can be
regarded as those from the dust particles with diameters smaller than 10 µm.
**4.3 Surface radiative effect (SRE) by aerosol-in-snow**
Figure 6 shows the spatial distribution of instantaneous surface radiative effect
(SRE) due to BC- and dust-induced snow albedo change, respectively, in winter
(December-January-February) and spring (March-April-May). Due to the decrease of
surface albedo, surface net shortwave radiation is increased. The spatial patterns of
SRE are determined collectively by both the amount of aerosol-in-snow and the
distribution of snowpack (snow depth and snow cover fraction). Finer-scale structures
of SRE in the Rocky Mountains and the adjacent regions are simulated by VR-CESM
with a higher horizontal resolution compared to previous simulations by
coarse-resolution GCMs (e.g., Flanner et al., 2009; Yasunari et al., 2015). The SRE is
generally above 0.2 W m$^{-2}$ over the mountains especially in the Greater Yellowstone
region and Southern Rockies. SRE can reach similar magnitudes on the southern
periphery of Northern Rockies and west side of the Greater Yellowstone region,
where higher near-surface atmospheric BC/dust concentrations and BC/dust-in-snow
mass mixing ratios are simulated (Figures 2 and 4). SRE is stronger in spring than in
winter for both BC and dust, which is consistent with previous studies (Flanner et al.,
2009; Yasunari et al., 2015). This is because of the stronger solar insolation and larger





albedo reduction due to snow aging and aerosol accumulation within snow in spring.
There are more dust emission and consequent dust transport and deposition in spring
than in winter, which may also partly contribute to the larger dust-induced SRE in
spring than in winter. For different aerosols, BC-induced SRE is somewhat larger than
dust-induced SRE in both winter and spring. BC-induced SRE is mostly below 1 W
$m^{-2}$ in winter, but reaches up to 2-5 W $m^{-2}$ in spring. Dust-induced SRE is mostly
below 0.5 W $m^{-2}$ in winter and increases to 1-5 W $m^{-2}$ in spring.

Compared to those in the Greater Yellowstone region and Southern Rockies,

SRE in the Northern Rockies is much smaller (mostly below 0.05 and 0.5 W $m^{-2}$ in
winter and spring), because of smaller aerosol-in-snow mass values in this region
(Figure 4). Note that BC-induced SRE is still significant (mostly around 0.2-2 W $m^{-2}$
and 2-5 W $m^{-2}$ in some local regions) in the northern Great Plains, eastern U.S.,
southern Canada, and eastern Canada. This was also shown in previous studies using
coarse-resolution GCMs (e.g., Flanner et al., 2009; Yasunari et al., 2015). In addition,
the model also simulates non-negligible dust-induced SRE (mostly around 0.05-0.2 W
$m^{-2}$ and up to 0.2-0.5 W $m^{-2}$ in some local regions) near the dust sources in southern
Canada and the northern Great Plains.

Figure 7 shows the monthly variations of SRE induced by BC and dust SDE in

the five regions (Northern Rockies, Greater Yellowstone region, Southern Rockies,
Eastern Snake River Plain, and Southwestern Wyoming). Table 2 gives the regional
averaged winter and spring SRE in these five regions. Consistent with the spatial



distributions shown in Figure 6, aerosol-induced SRE averaged in Northern Rockies is
about half to one-fourth of those in Greater Yellowstone region and Southern Rockies.
Compared to that in winter, SRE is much larger in spring, which is a result of aerosol
accumulation in snow and relatively strong solar insolation. The peaks of SRE occur
in April-May in the three mountainous regions (Northern Rockies, Greater
Yellowstone region, and Southern Rockies), which corresponds to the onset of
snowmelt after the peaks of snow water equivalent (early-to-middle April; see figure
11 of Wu et al. (2017)). In the Eastern Snake River Basin and Southwestern Wyoming,
the peaks of SRE occur in March, which is different from the three mountainous
regions because the snowmelt period begins earlier (in February-to-March) in these
two regions (section 4.4). Regional mean total SRE in spring induced by BC and dust
can reach up to 1.58-1.7 W m$^2$ with peaks around 2.0 W m$^{-2}$ in Greater Yellowstone
region and Southern Rockies. In Eastern Snake River Plain and Southwestern
Wyoming, regional mean total SRE in winter and spring is around 0.4-0.6 W m$^{-2}$ and
0.7-0.8 W m$^{-2}$, respectively. For the contribution of different aerosols, BC-induced
springtime SRE is larger than dust-induced SRE in the five regions. Despite this,
dust-induced springtime SRE can still contribute to about 20-30% of total springtime
SRE in the northern part of Rocky Mountains (Northern Rockies, Greater
Yellowstone region and Eastern Snake River Plain). In the southern part of Rocky
Mountains (Southwestern Wyoming and Southern Rockies), dust-induced springtime
SRE contributes more significantly (about 30-40%) to total springtime SRE.





**4.4 Impacts of aerosol SDE on the surface temperature and snowpack**

Figure 8 shows surface air temperature, snow water equivalent, and snow

cover fraction changes due to the aerosol SDE in winter and spring, respectively.

Snow water equivalent is defined as the amount of water contained within the

snowpack, measured by kg m$^{-2}$ which is equivalent to mm after dividing the density

of water (1000 kg m$^{-3}$). Snow cover fraction is defined as the fraction of surface area

covered by snow. These changes are derived from the difference between two

simulations (CTL and NoSDE). The crosses in the figure denote the regions where

changes are statistically significant at the 0.1 level. Although SRE is largest over the

mountains, surface air temperature change is largest around the mountains, such as

over the Eastern Snake River Plain, Northern Utah, and Central-Southwestern

Wyoming, where surface air temperatures are increased by around 0.5-2 °C due to the

aerosol SDE. The large surface air temperature increase corresponds well to the

significant reductions of snow water equivalent (by 2-50 mm) and snow cover

fraction (by 5-20%) in these regions. This indicates a pronounced positive feedback

between snow albedo, radiation, and surface temperature around the mountains,

where snow water equivalent values are relatively lower and snow cover fractions are

smaller than those over the mountains. The positive feedback amplifies the surface

warming and snow melting, as was also found in a previous study using the Weather

Research and Forecasting (WRF) model (Qian et al., 2009). We note, however, both

snow water equivalent and snow cover fraction are larger over the mountains. For

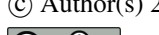



example, winter and spring snow cover fraction is mostly above 70% on the high
mountains (see Figure 7 of Wu et al. (2017)). This suggests that snow on the high
mountains is less susceptible to the aerosol SDE. Another reason for the smaller
change of surface air temperature over the mountains is that snow water equivalent
and snow cover fraction are increased (especially in the Northern Rockies and Greater
Yellowstone region) due to the increase of snowfall in these regions (Figure 9f),
which cancels out the reduction of snow water equivalent resulting from aerosol SDE.
The increase of snowfall is due to enhanced water vapor transport from the Pacific
Ocean (Figure not shown), which is likely related to the large-scale circulation change
due to aerosol SDE. Note that increases of snow water equivalent and snow cover
fraction in the Northern Rockies and Greater Yellowstone region due to aerosol SDE
do not pass the significant test at 0.1 level because of the large interannual variability
in these regions.

Table 2 gives the winter and spring surface air temperature changes due to

LAAs in snow averaged over the five regions. Seasonal mean surface air temperature
change is around 0.9-1.1 °C in the Eastern Snake River Plain in winter and spring,
while this change is around 0.003-0.17 °C (winter) and around 0.3-0.5 °C (spring) in
the mountainous regions (Northern Rockies, Greater Yellowstone region, and
Southern Rockies). In Table 2, we also show the efficacy of snow albedo forcing,
which is defined as the ratio of surface air temperature change to SRE. The efficacy is
mostly around 0.1-0.5 in the three mountainous regions, but it is 1.3-2.2 in the Eastern



Snake River Plain and Southwestern Wyoming. This indicates that stronger snow
albedo feedbacks exist in the latter two regions.

Figures 10-11 show the monthly evolution of regional mean surface air

temperature, snow water equivalent, and snow cover fraction, and their changes due
to aerosol SDE in the Eastern Snake River Plain and Southwestern Wyoming,
respectively. Monthly variations of surface air temperature, snow water equivalent,
and snow cover fraction are similar between the two regions: lowest surface air
temperature and largest snow cover fraction in January, and highest snow water
equivalent in February-March. Significant changes of these variables from the aerosol
SDE occur in both regions. The largest surface air temperature increase is 1.5 °C in
the Eastern Snake River Plain and 1.6 °C in Southwestern Wyoming, occurring in
April and December, respectively. In the Eastern Snake River Plain (Southwestern
Wyoming), aerosol SDE leads to the reduction of snow water equivalent by 6-28 mm
(6-16 mm) and snow cover fraction by 5-15% (6-19%) from December to March,
which corresponds to a fraction of 18-33% (20-37%) and 7-28% (8-28%),
respectively, with respect to the NoSDE simulation results. In April (late snowmelt
period) when snow water equivalent and snow cover fraction are both relatively
small, the aerosol SDE is more significant, which reduces snow water equivalent
values and snow cover fractions by about half.
**4.5 Runoff change induced by aerosol SDE**



In the model, the runoff includes the surface runoff and sub-surface runoff.
Runoff is mainly from the rainfall and snowmelt. The change of rainfall is shown in
Figures 9c-9d, and the snowmelt change is shown in Figure 12. Aerosol SDE
increases the snowmelt by 0.1-2 mm/day in the mountains during the snow
accumulation and early snowmelt period (in autumn, winter and spring). In the late
snowmelt period, aerosol SDE reduces the snowmelt due to less snowpack available
for melting in the plains (in spring) and in the mountains (in summer). Note that
snowmelt is slightly reduced by the aerosol SDE in autumn in the Southern Rockies,
which is a result of less snowpack for melting due to the reduced snowfall in this
region (Figure not shown).
Because of the change in rainfall and snowmelt due to aerosol SDE, surface
runoff changes too. Figure 13 shows the runoff change induced by aerosol SDE in
four seasons. In winter, runoff is barely modified by the aerosol SDE in the Rocky
Mountains, except in the Northern Rockies where runoff is increased by 0.1-2 mm
day$^{-1}$, associated with increased rainfall (Figure 9c) and increased snowmelt (Figure
12a). In spring, runoff changes the most compared to all of the other seasons, with the
runoff increased by up to 0.5-2 mm day$^{-1}$ in the mountainous regions. This is mainly
due to the increase of snowmelt resulting from surface warming (Figure 12b) as well
as due to more snow available for melt resulting from snowfall increase (Figure 9f).
The changes in runoff are statistically significant at 0.1 level in most of the
mountainous regions in spring. Absolute runoff increases are stronger in the Northern



Rockies and Greater Yellowstone regions than in the Southern Rockies in terms of the
area and magnitude, probably due to the smaller snow water equivalent in Southern
Rockies (Wu et al., 2017). As more snowmelt occurs in spring, less snowpack is
available for melt in summer and thus surface runoff is reduced by about 0.1-1 mm
day$^{-1}$. There is little runoff change in autumn, as there is less runoff generated from
rainfall and snowmelt than in other seasons. Overall, BC and dust residing in snow
accelerate the hydrologic cycles by increasing the runoff in spring and reducing the
runoff in summer. Surface warming also increases the ratio of rainfall to total
precipitation, which can accelerate the generation of runoff. Note that in some regions
of the plains, such as the central-eastern Montana, Southwestern Wyoming, and Snake
River Basin, the snowmelt changes by 0.1-1 mm/day due to the aerosol SDE, but the
runoff changes little. This is because the water generated from snowmelt is mainly
stored in soil or transformed into evapotranspiration in these regions. Also note that
there are statistically significant increases of runoff in the southern Great Plains in
spring, but the change is small (around 0.05-0.1 mm/day; Figure 13b). This change is
a result of slight increases in both rainfall (Figure 9d) and snowmelt (Figure 12b).
Figure 14 shows the monthly evolution of runoff and its change due to the
aerosol SDE in the three mountainous regions (Northern Rockies, Greater
Yellowstone region, and Southern Rockies). In the three regions, runoff peaks in the
late spring and early summer (in May in Northern Rockies and Southern Rockies, and
in June in Greater Yellowstone region) when snow melting progresses after the peak



of snow water equivalent in early-to-middle April (Wu et al., 2017). This indicates the
significant contribution of snowmelt to runoff. Overall, runoff changes are larger in
the Northern Rockies and Greater Yellowstone region than in Southern Rockies,
which is consistent with the spatial distribution of runoff changes shown in Figure 13.
Runoff is significantly increased in spring and decreased in June and July, indicating
the acceleration of the local hydrologic cycle by aerosol SDE. In the Northern
Rockies, runoff is also increased from October to March but in much smaller
magnitudes (below 0.2 mm day$^{-1}$) compared to April and May. In April (May), runoff
is increased by 0.39 (0.56), 0.22 (1.00), and 0.17 (0.15) mm day$^{-1}$ in the the Northern
Rockies, Greater Yellowstone region, and Southern Rockies, respectively. This
increase contributes to 26% (13%), 42% (27%), and 29% (7%) of the runoff from the
NoSDE simulation in April (May) for the three regions, respectively. The reduction of
runoff in June is relatively small (0.06 and 0.11 mm day$^{-1}$, respectively) in the
Northern Rockies and Greater Yellowstone, only accounting for 2% of runoff from
the NoSDE simulation. However, it reaches up to 0.18 mm day$^{-1}$ in the the Southern
Rockies, which accounts for 15% of runoff. In addition, due to the reduction of snow
available for melting in later month (i.e., July), the runoff is further reduced. With
respect to the relative smaller runoff in July than in previous months, aerosol SDE is
more significant, which can reduce the runoff by 0.04 (8%), 0.17 (23%), and 0.06 mm
day$^{-1}$ (16%) in the three regions, respectively. Note that due to increase of



precipitation, the annual mean runoff is increased by 0.12 (12%), 0.09 (10%), and
0.01 mm day$^{-1}$ (2%) in these three regions, respectively.
**5.  Conclusions**

In this study, we use VR-CESM to quantify the impacts of LAAs (BC and

dust) deposition on the snowpack and hydrologic cycles in the Rocky Mountains. Our
previous study has shown that VR-CESM reproduces reasonably the spatial
distributions and seasonal evolution of snowpack in the Rocky Mountains (Wu et al.,
2017). Here we show that the model also reproduces observed spatial distributions of
near-surface BC and dust concentrations in the western U.S. compared to IMPROVE
observations. The magnitude of simulated near-surface dust concentrations is
comparable to observations, while that of simulated near-surface BC concentrations is
mostly underestimated, especially in the Rocky Mountain region. The
underestimation of near-surface BC concentrations may be due to the absence of local
sources for BC emissions used in the model. Simulated aerosol-in-snow
concentrations are closely related to the distributions of both snowpack and
near-surface atmospheric aerosol concentrations. Simulated BC-in-snow
concentrations are mostly comparable to the observations with the magnitude between
2 and 30 ng g$^{-1}$, although significant underestimations (by as much as a factor of 10)
were found at two stations.

Due to the deposition of LAAs on snow cover, surface net shortwave radiation

is increased. Regional averaged SRE induced by LAAs in snow is 0.1-0.5 W m$^{-2}$ in



winter in the three mountainous regions (Northern Rockies, Greater Yellowstone
region, and Southern Rockies) and 0.4-0.6 W m$^{-2}$ in the two regions around the
mountains (Eastern Snake River Plain and Southwestern Wyoming). SRE is much
larger in spring and reaches up to 0.6-1.7 W m$^{-2}$ in these five regions (Table 2). Dust
contributes 21-43% to the total SRE induced by LAAs in snow in spring, indicating
the important role of dust residing in snow. Of the five regions, dust contributes the
most (43%) to the total SRE in the Southern Rockies. This is not unexpected as this
region is close to dust sources in the Colorado Plateau.

As a result of SRE induced by LAAs in snow, surface air temperature

increases in most of the Rocky Mountain region. The surface air temperature increase
is largest over the Eastern Snake River Plain and Southwestern Wyoming, with winter
and spring surface air temperature increased by 0.9-1.1°C. Significant reductions of
snow water equivalent (by 2-50 mm) and snow cover fraction (by 5-20%) occur in
these two regions, indicating a strong positive snow-albedo feedback there.

Aerosol SDE accelerates the hydrologic cycle in the mountainous regions. In

April and May, monthly mean runoff is increased by 7%-42% in the three
mountainous regions (Northern Rockies, Greater Yellowstone region, Southern
Rockies). This is because of the accelerated snowmelt resulting from surface warming
as well as the increased snowfall resulting from enhanced water vapor transport from
the Pacific Ocean. The enhanced water vapor transport may be related to large-scale
circulation changes. In the later stage of snowmelt, monthly runoff is reduced by 2-15%



in June and 8-23% in July in the three mountainous regions. In particular, aerosol
SDE leads to a reduction of total runoff by about 15% in June and July in the
Southern Rockies. This highlights the important role of aerosol SDE in modulating
the hydrologic cycle in the mountainous regions.

We note that VR-CESM still underestimates the near-surface BC

concentrations, however, reproduces observed magnitudes of BC-in-snow
concentrations for most stations. For dust in snow, the model used in this study only
accounts for dust particles smaller than 10 μm, while observations made by Reynolds
et al. (2016) suggest that most airborne dust concentrations are characterized by dust
particles with diameters larger than 10 μm in the Utah-Colorado region. Therefore,
our simulations may significantly underestimate the impacts of dust in snow. Actually,
our simulations suggest SRE induced by dust-in-snow can reach up to 2-5 W m$^{-2}$ in
the Southern Rockies, which is nearly an order of magnitude smaller than values in
Painter et al. (2007) based on observed dust-in-snow particles. Future observations of
LAAs in snow, particularly for the temporal evolution of LAAs in different snow
layers, as well as detailed size distribution measurements of dust particles in snow
will help reduce the uncertainties in the model quantification of the impacts of LAAs
in snow.

Although the uncertainties still exist, our results show LAAs in snow can

significantly affect the snowpack and consequent hydrologic cycle in the Rocky
Mountains. Previous studies have demonstrated that snowpack on the Rocky



Mountains has declined significantly in the second half of 20th century (e.g.,
Pederson et al., 2011). The role of LAAs in this decrease of snowpack is still
unknown. It would be interesting to investigate the role of LAAs and compare it with
those of other climate factors (such as natural climate variability and greenhouse gas
concentrations). Moreover, BC and dust emissions may also be subject to changes in
the future. Therefore, for better projections of future changes in Rocky Mountain
snowpack, the impacts of LAAs in snow under future emissions scenarios also need to
be taken into account.

**Acknowledgement**
This research is supported by the University of Wyoming Tier-1 Engineering
Initiative for High-Performance Computational Science and Technology funded by
the State of Wyoming. Z. Lin was jointly supported by the Special Scientific Research
Fund of the Meteorological Public Welfare Profession of China (grant
GYHY01406021), National Key Research and Development Program of China (grant
2016YFC0402702), and the National Natural Science Foundation of China (grant
41575095). We thank the team for maintaining the Interagency Monitoring of
PROtected Visual Environments (IMPROVE) network and making the observation
dataset available to use (http://vista.cira.colostate.edu/Improve/improve-data/). We
also thank Dr. Sarah Doherty from the University of Washington and Dr. S.
McKenzie Skiles from Jet Propulsion Laboratory, California Institute of Technology



for providing the observations of absorbing aerosols in snow and helpful suggestions
on use of the data. We thank Alan M. Rhoades and Paul A. Ullrich from University of
California, Davis as well as Colin M. Zarzycki from NCAR for helpful discussions
during this study. We would like to acknowledge the use of computational resources
by conducting the model simulations (ark:/85065/d7wd3xhc) at the NCAR-Wyoming
Supercomputing Center provided by the NSF and the State of Wyoming, and
supported by NCAR's Computational and Information Systems Laboratory. The
simulation results can be obtained by contacting the corresponding author X. Liu
(xliu6@uwyo.edu).

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



**Table 1.** Observations of BC mass concentration in snow column ($C_{BC}$, ng g$^{-1}$, i.e., ng
gram BC per g snow) in the Rocky Mountain region compiled from previously
published literature.

| No. | Latitude (°N) | Longitude (°W) | Elevation (m) | Date sampled | $C_{BC}$ (ng g$^{-1}$)[a] | Source |
|---|---|---|---|---|---|---|
| 1 | 40.9014 | 115.8910 | 1949 | 2/1/13 | 7.6 | Site 8 of Doherty et al. (2014) |
| 2 | 42.2767 | 116.0115 | 1772 | 2/1/13 | 5.6 | Site 9 of Doherty et al. (2014) |
| 3 | 43.3495 | 115.3968 | 1538 | 2/3/13 | 6.0 | Site 10 of Doherty et al. (2014) |
| 4 | 43.5927 | 113.5894 | 1942 | 2/3/13 | 5.8 | Site 11 of Doherty et al. (2014) |
| 5 | 43.4010 | 111.2053 | 1727 | 2/4/13 | 6.8 | Site 12 of Doherty et al. (2014) |
| 6 | 42.9357 | 109.8576 | 2274 | 2/4/13 | 6.3 | Site 13 of Doherty et al. (2014) |
| 7 | 41.7297 | 109.3668 | 2223 | 2/5/13 | 29.1 | Site 14 of Doherty et al. (2014) |
| 8 | 40.7464 | 109.4776 | 2583 | 2/5/13 | 9.3 | Site 15 of Doherty et al. (2014) |
| 9 | 40.1316 | 109.4711 | 1538 | 2/7/13 | 14.3 | Site 16 of Doherty et al. (2014) |
| 10 | 40.4929 | 107.8994 | 1962 | 2/8/13 | 9.5 | Site 17 of Doherty et al. (2014) |
| 11 | 40.6695 | 106.4158 | 2512 | 2/9/13 | 11.4 | Site 18 of Doherty et al. (2014) |
| 12 | 48.2318 | 105.0949 | 648 | 2/17/13 | 10.9 | Site 24 of Doherty et al. (2014) |
| 13 | 44.9475 | 116.0813 | 1528 | January to March, 2014 | 9.8 (5.4) | Site McCall of Doherty et al. (2016) |
| 14 | 44.4224 | 115.9899 | 1450 | February to March, 2014 | 13.3 (9.5) | Site Cascade Valley of Doherty et al. (2016) |
| 15 | 44.0949 | 115.9771 | 960 | February to March, 2014 | 14.9 (8.9) | Site Garden Valley of Doherty et al. (2016) |
| 16 | 40.143 | 109.467 | 1620 | February to March, 2013 and 2014 | 33.6 (25.4) | Site Vernal of Doherty et al. (2016) |
| 17 | 37.9069 | 107.7113 | 3368 | March to May, 2013 | 9.1 (7.9) | At Senator Beck Basin Study Area (SBBSA) (Skiles and Painter, 2016b) |

[a]: If multi-measurements of $C_{BC}$ are made during the observation period, the mean $C_{BC}$ is given
with the standard deviation of $C_{BC}$ shown in parenthesis next to the mean $C_{BC}$.






**Table 2.** Winter (December-January-February) and spring (March-April-May) mean
surface shortwave radiative effect (SRE; W m$^{-2}$) due to BC alone, dust alone and BC
and dust together in snow, as well as surface air temperature (SAT; ºC) change and
the efficacy of SRE in SAT change in the five regions (see Figure 1c). Note that SRE
induced by BC and dust together is slightly larger than the sum of SRE induced by
BC and SRE by dust separately.

| Season | SRE by BC[a] | SRE by dust[a] | SRE by BC & dust | SAT change | Efficacy[b] |
|---|---|---|---|---|---|
| Northern Rockies | | | | | |
| Winter | 0.13 (92%) | 0.01 (8%) | 0.14 | 0.08 | 0.57 |
| Spring | 0.42 (79%) | 0.11 (21%) | 0.57 | 0.32 | 0.56 |
| Greater Yellowstone region | | | | | |
| Winter | 0.24 (88%) | 0.03 (12%) | 0.28 | 0.004 | 0.014 |
| Spring | 1.11 (71%) | 0.45 (29%) | 1.70 | 0.50 | 0.29 |
| Southern Rockies | | | | | |
| Winter | 0.36 (77%) | 0.11 (23%) | 0.50 | 0.17 | 0.34 |
| Spring | 0.79 (58%) | 0.58 (42%) | 1.58 | 0.30 | 0.19 |
| Eastern Snake River Plain | | | | | |
| Winter | 0.50 (84%) | 0.09 (16%) | 0.62 | 0.93 | 1.5 |
| Spring | 0.54 (73%) | 0.20 (27%) | 0.80 | 1.13 | 1.41 |
| Southwestern Wyoming | | | | | |
| Winter | 0.33 (81%) | 0.08 (19%) | 0.43 | 0.93 | 2.16 |
| Spring | 0.43 (67%) | 0.22 (33%) | 0.70 | 0.90 | 1.29 |

[a]: The fraction of SRE by BC (dust) to the sum of SRE by BC and SRE by dust is given in
parenthesis next to SRE by BC (dust).
[b]: Efficacy of snow/ice albedo forcing (ºC increase per 1 W m$^{-2}$) is defined as the ratio of SAT
change to SRE.





**Figure captions:**

**Figure 1.** (a) Model meshes for variable resolution (uniform 1° with refined 0.125° in
the Rocky Mountains) used in VR-CESM. Note that each element shown contains
additional 3×3 collocation gridcells. (b) Terrain height (m) in the western US with the
refined region at a resolution of 0.125º surrounded by dashed lines. (c) Five regions
identified for the analysis in this study, including three mountainous region (1,
Northern Rockies; 2, Greater Yellowstone region; 3, Southern Rockies) and two
regions in the plains around the mountains (4, Eastern Snake River Plain; 5,
Southwestern Wyoming).
**Figure 2.** Spatial distribution of cold season (winter and spring) mean (a) BC
emission flux and (b) near-surface BC concentration from the VR-CESM simulation;
(c) and (d) for dust emission flux and near-surface dust concentration, respectively.
Also shown are the IMPROVE stations (blue open circle) selected for model
validation, with the size of the circles from small to large indicating the magnitude of
observed near-surface BC/dust concentrations. The black rectangles in (b) and (d)
denotes the five regions (A, West Coast; B, Rocky Mountains; C, Utah and Nevada;
D, Southwestern US; E, Great Plain), which will be used to classify the stations in
Figure 3. Note that units for BC and dust concentrations are ng m$^{-3}$ and μg m$^{-3}$,
respectively.
**Figure 3.** Comparison of cold season mean near-surface (a) BC and (b) dust
concentrations at IMPROVE stations from VR-CESM simulation and IMPROVE
observations. Also given are the mean results at all the stations from simulation and
observations and their correlation coefficient (*R*). The 1:1 (solid) and 1:5/5:1 (dash)
lines are plotted for reference.
**Figure 4.** Winter (December-January-February (DJF); left) and spring
(March-April-May (MAM); right) mean BC (upper row) and dust (bottom row) mass
mixing ratios in snow column. Also shown are the stations for observations of BC
mass in snow column (a, b) and for observations of dust mass in snow column at
Senator Beck Basin Study Area (SBBSA) in the San Juan Mountains (c, d). Note that
the units for BC and dust mass mixing ratios are given in different units, i.e., ng g$^{-1}$
and μg g$^{-1}$, respectively.
**Figure 5.** Comparison of BC mass concentrations in the snow column ($C_{BC}$) at the 17
sites (see Table 1) from VR-CESM simulations and observations with the error bars



denoting the corresponding standard deviations. The observations are compiled from
the previously published studies (Table 1). If multiple observations are recorded at a
certain site, the observed standard deviations are calculated from these multiple
observations (section 3). Simulated BC mass concentration in the snow column and
its standard deviation are calculated from the 25-year mean and standard deviation of
simulation in the same month as the observations (section 3).
**Figure 6.** Winter (December-January-February (DJF), left) and spring
(March-April-May (MAM), right) mean surface shortwave radiative effect (SRE, W
$m^{-2}$) induced by BC (top) and dust (bottom).
**Figure 7.** Monthly variations of surface radiative effect (SRE; W $m^{-2}$) during the
water year (October 1st to September 30th) averaged over the Northern Rockies,
Greater Yellowstone region, Southern Rockies, Eastern Snake River Basin, and
Southwestern Wyoming, respectively.
**Figure 8.** Changes in surface air temperature (upper row; ℃), snow water equivalent
(bottom row; mm), and snow cover fraction (bottom row; %) in winter (left) and
spring (right) induced by BC- and dust-in-snow. The crosses denote the regions where
changes are statistically significant at 0.1 level.
**Figure 9.** As Figure 8, but for total precipitation change (top), rainfall change (center),
and snowfall change (bottom). The unit is mm $day^{-1}$.
**Figure 10.** Seasonal evolution of (a) surface air temperature, (c) snow water
equivalent, and (e) snow cover fraction and their changes due to SDE (b, d, and f)
averaged over the Eastern Snake River Plain.
**Figure 11.** As Figure 10, but for Southwestern Wyoming.
**Figure 12.** Snowmelt change (mm $day^{-1}$) due to SDE of BC and dust in four seasons:
(a) December-January-February (DJF), (b) March-April-May (MAM), (c)
June-July-August (JJA), and (d) September-October-November (SON). The crosses
denote the regions where changes induced by SDE are statistically significant at 0.1
level.
**Figure 13.** As Figure 12, but for runoff change (mm $day^{-1}$).
**Figure 14.** Seasonal evolution of runoff (left) and their change (right) in the Northern
Rockies (top), the Greater Yellowstone region (center), and Southern Rockies
(bottom). The unit is mm $day^{-1}$.







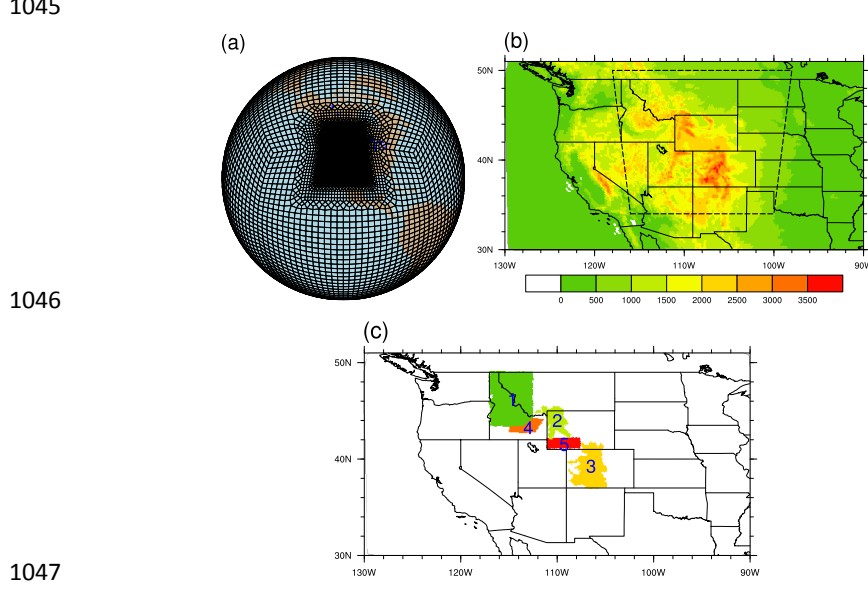




**Figure 1.** (a) Model meshes for variable resolution (uniform 1° with refined 0.125° in
the Rocky Mountains) used in VR-CESM. Note that each element shown contains
additional 3×3 collocation gridcells. (b) Terrain height (m) in the western US with the
refined region at a resolution of 0.125º surrounded by dashed lines. (c) Five regions
identified for the analysis in this study, including three mountainous region (1,
Northern Rockies; 2, Greater Yellowstone region; 3, Southern Rockies) and two
regions in the plains around the mountains (4, Eastern Snake River Plain; 5,
Southwestern Wyoming).







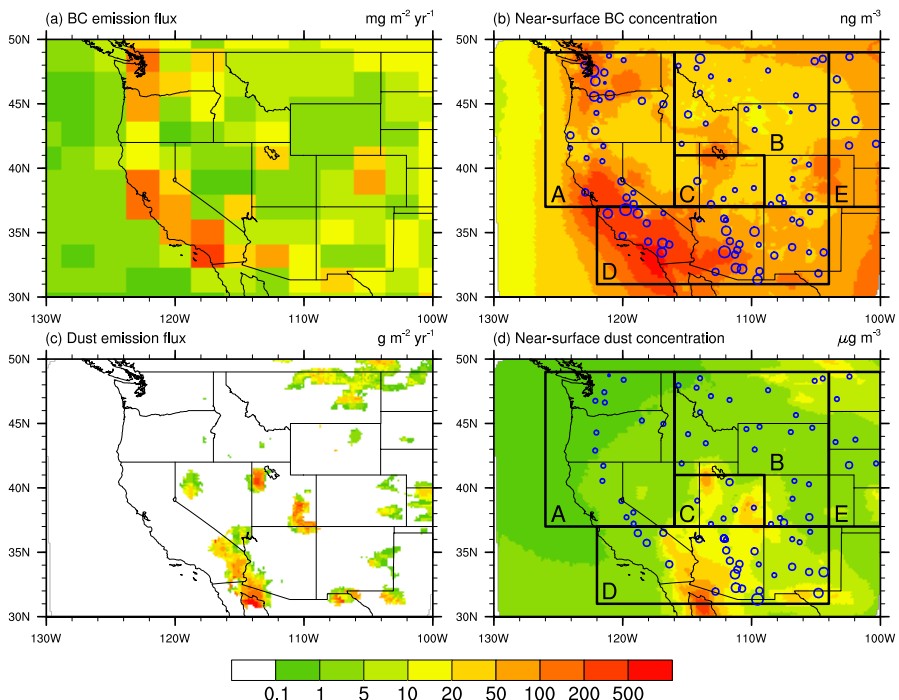


**Figure 2.** Spatial distribution of cold season (winter and spring) mean (a) BC

emission flux and (b) near-surface BC concentration from the VR-CESM simulation;

(c) and (d) for dust emission flux and near-surface dust concentration, respectively.

Also shown are the IMPROVE stations (blue open circle) selected for model

validation, with the size of the circles from small to large indicating the magnitude of

observed near-surface BC/dust concentrations. The black rectangles in (b) and (d)

denotes the five regions (A, West Coast; B, Rocky Mountains; C, Utah and Nevada;

D, Southwestern US; E, Great Plain), which will be used to classify the stations in

Figure 3. Note that units for BC and dust concentrations are ng m$^{-3}$ and μg m$^{-3}$,

respectively.









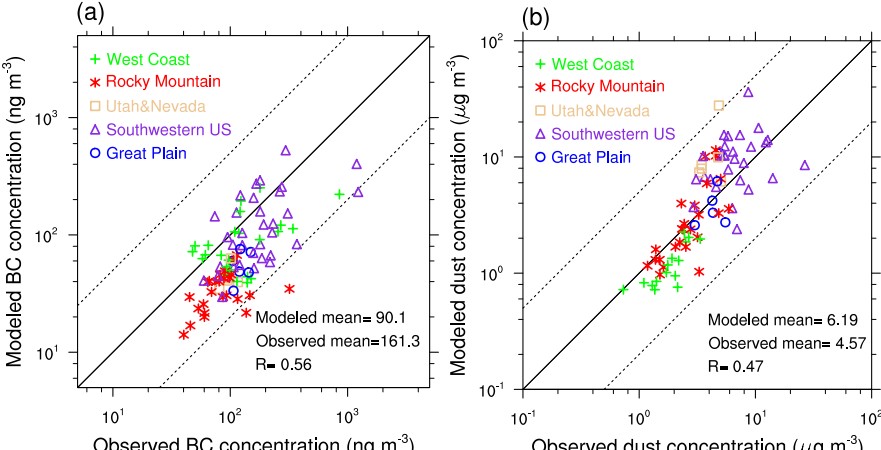


**Figure 3.** Comparison of cold season mean near-surface (a) BC and (b) dust

concentrations at IMPROVE stations from VR-CESM simulation and IMPROVE

observations. Also given are the mean results at all the stations from simulation and

observations and their correlation coefficient (*R*). The 1:1 (solid) and 1:5/5:1 (dash)

lines are plotted for reference.

1081

1082





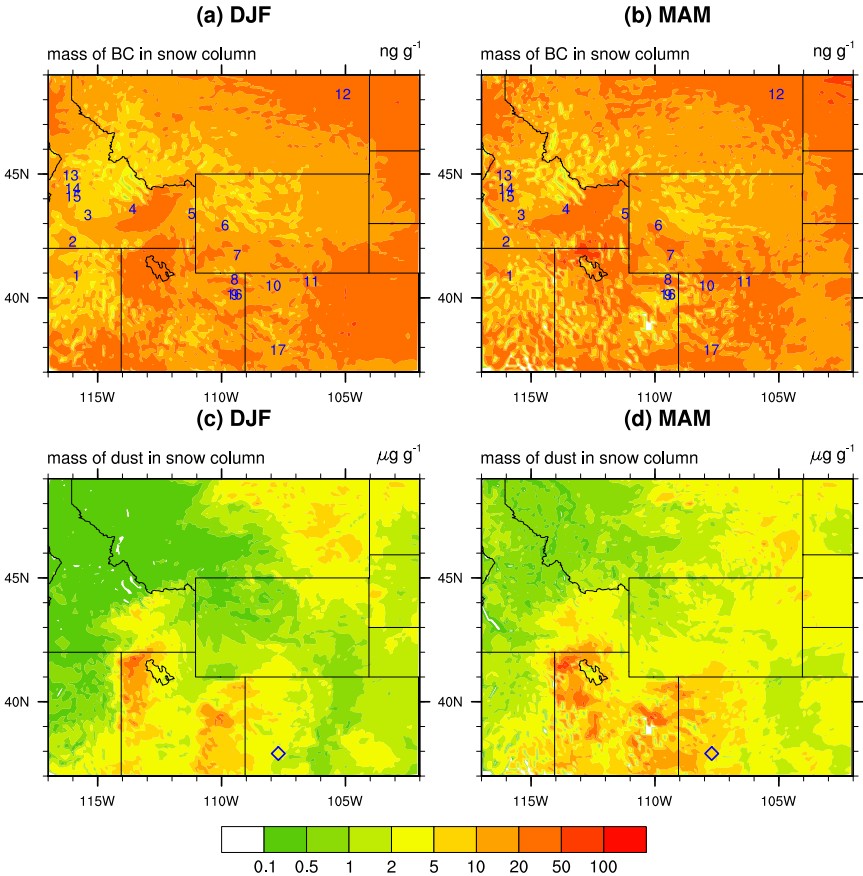

**Figure 4.** Winter (December-January-February (DJF); left) and spring
(March-April-May (MAM); right) mean BC (upper row) and dust (bottom row) mass
mixing ratios in snow column. Also shown are the stations for observations of BC
mass in snow column (a, b) and for observations of dust mass in snow column at
Senator Beck Basin Study Area (SBBSA) in the San Juan Mountains (c, d). Note that
the units for BC and dust mass mixing ratios are given in different units, i.e., ng g$^{-1}$
and μg g$^{-1}$, respectively.





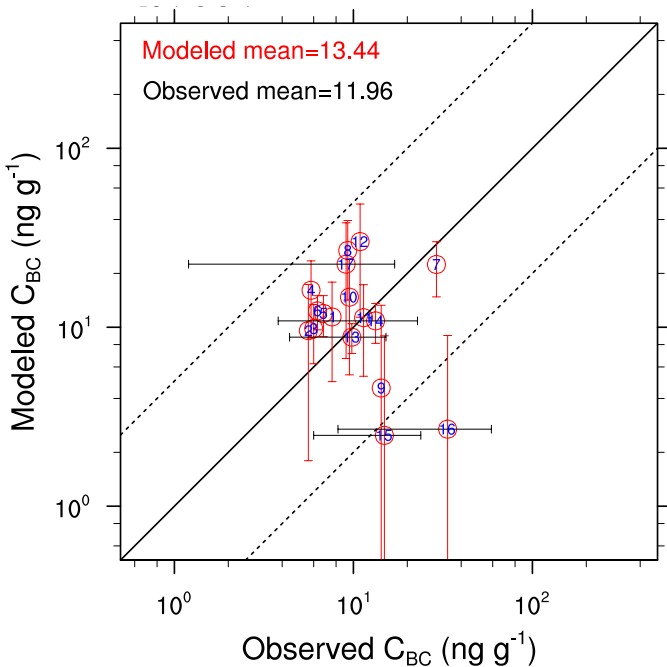

**Figure 5.** Comparison of BC mass concentrations in the snow column ($C_{BC}$) at the 17

sites (see Table 1) from VR-CESM simulations and observations with the error bars

denoting the corresponding standard deviations. The observations are compiled from

the previously published studies (Table 1). If multiple observations are recorded at a

certain site, the observed standard deviations are calculated from these multiple

observations (section 3). Simulated BC mass concentration in the snow column and

its standard deviation are calculated from the 25-year mean and standard deviation of

simulation in the same month as the observations (section 3).





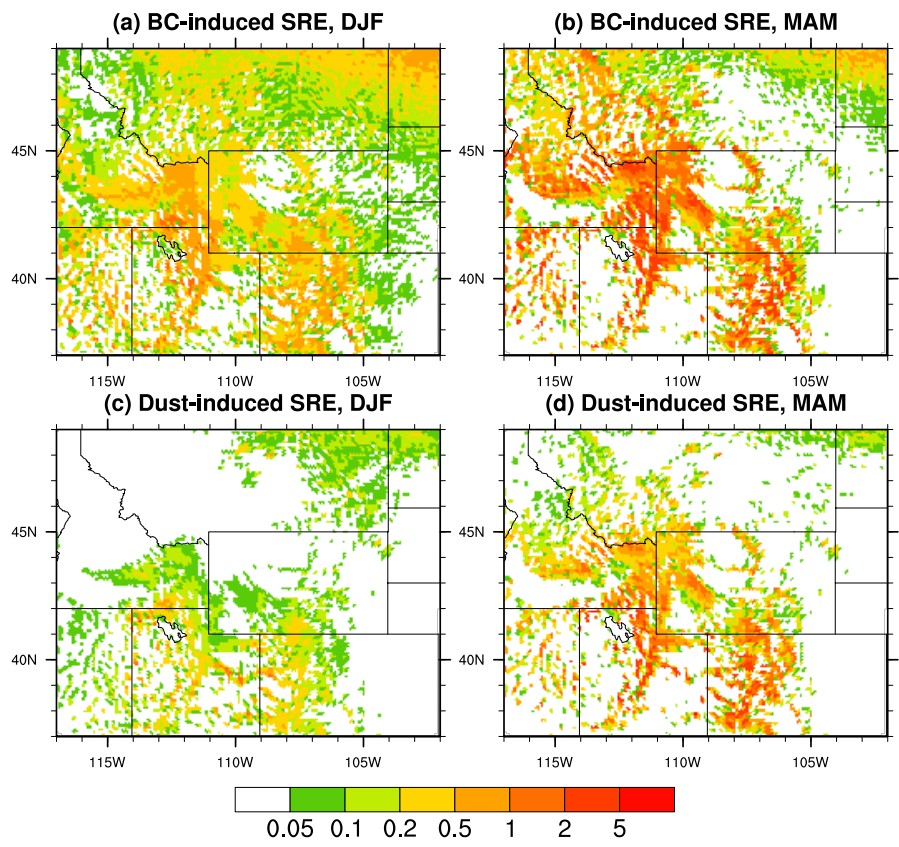


**Figure 6.** Winter (December-January-February (DJF), left) and spring

(March-April-May (MAM), right) mean surface shortwave radiative effect (SRE, W

m$^{-2}$) induced by BC (top) and dust (bottom).







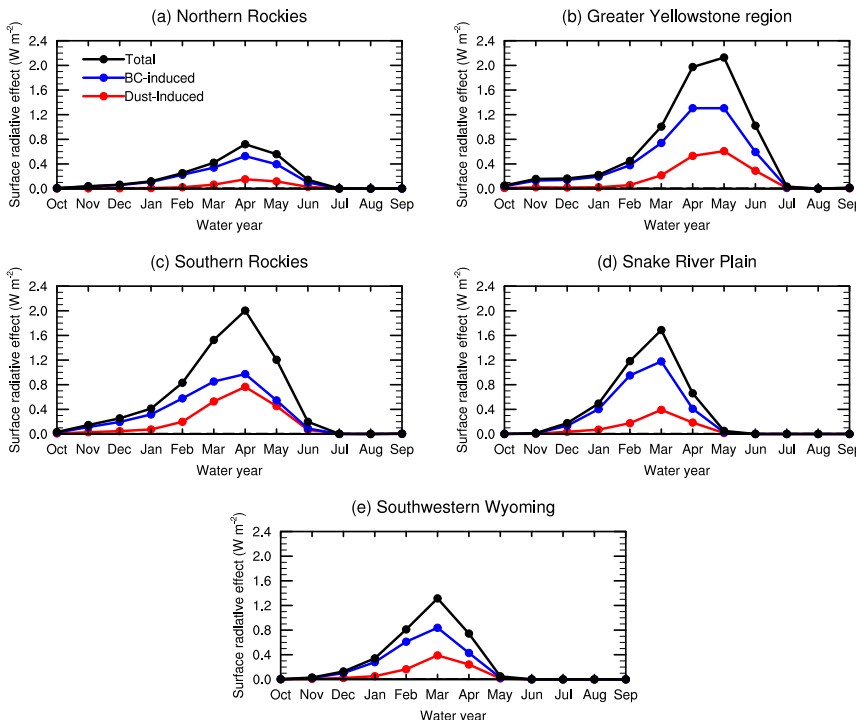


**Figure 7.** Monthly variations of surface radiative effect (SRE; W m$^{-2}$) during the
water year (October 1st to September 30th) averaged over the Northern Rockies,
Greater Yellowstone region, Southern Rockies, Eastern Snake River Basin, and
Southwestern Wyoming, respectively.

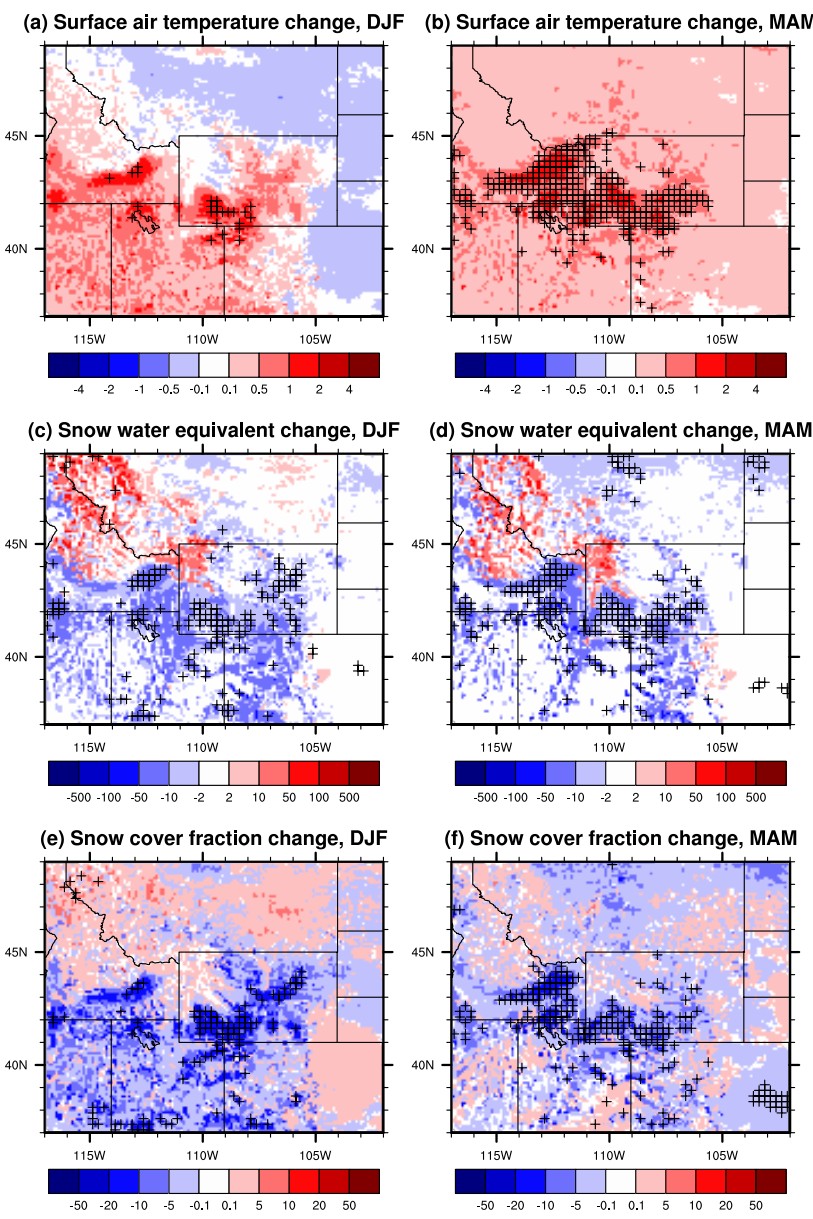

**Figure 8.** Changes in surface air temperature (upper row; ºC), snow water equivalent
(bottom row; mm), and snow cover fraction (bottom row; %) in winter (left) and
spring (right) induced by BC- and dust-in-snow. The crosses denote the regions where
changes are statistically significant at 0.1 level.





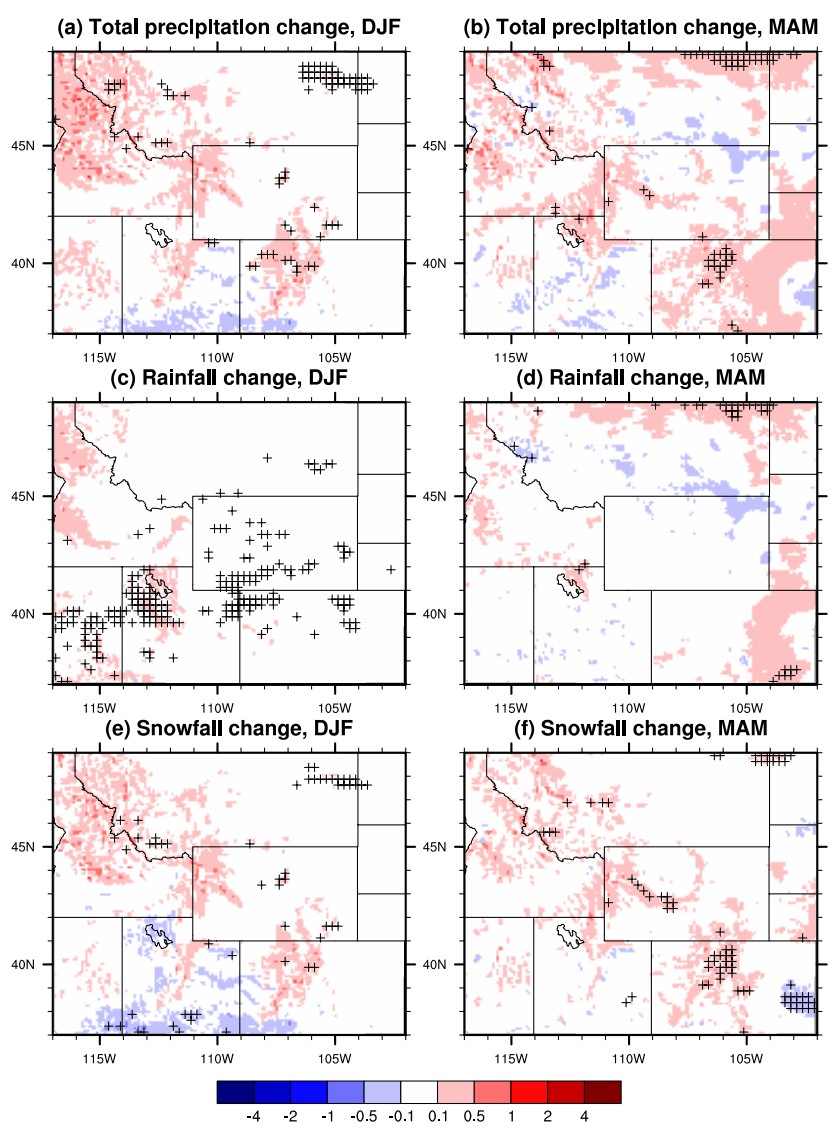

**Figure 9.** As Figure 8, but for total precipitation change (top), rainfall change (center), and snowfall change (bottom). The unit is mm day$^{-1}$.



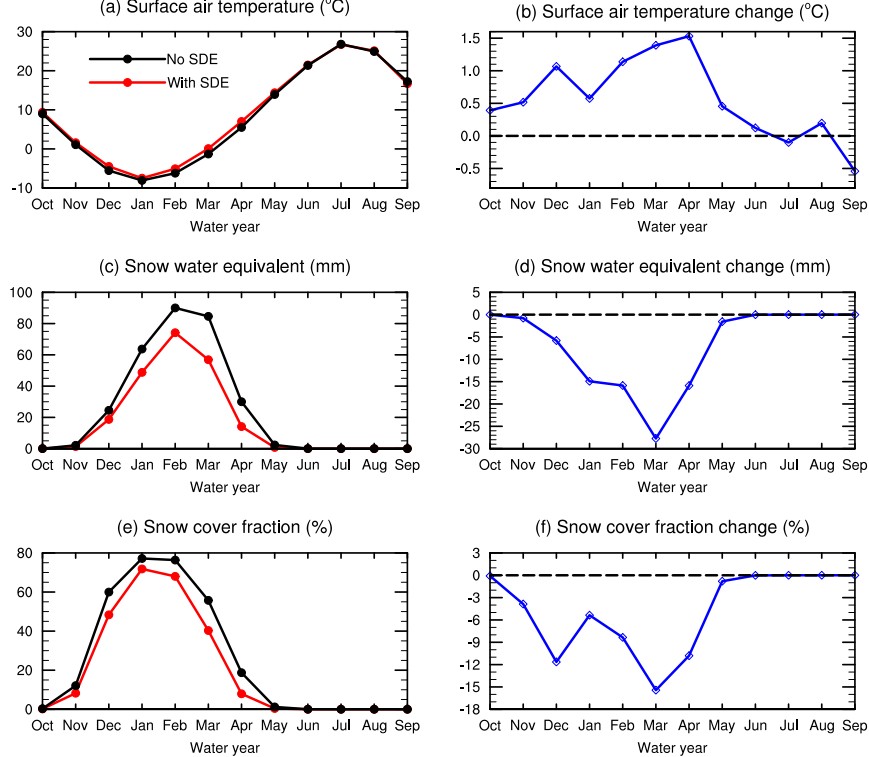

**Figure 10.** Seasonal evolution of (a) surface air temperature, (c) snow water

equivalent, and (e) snow cover fraction and their changes due to SDE (b, d, and f)

averaged over the Eastern Snake River Plain.





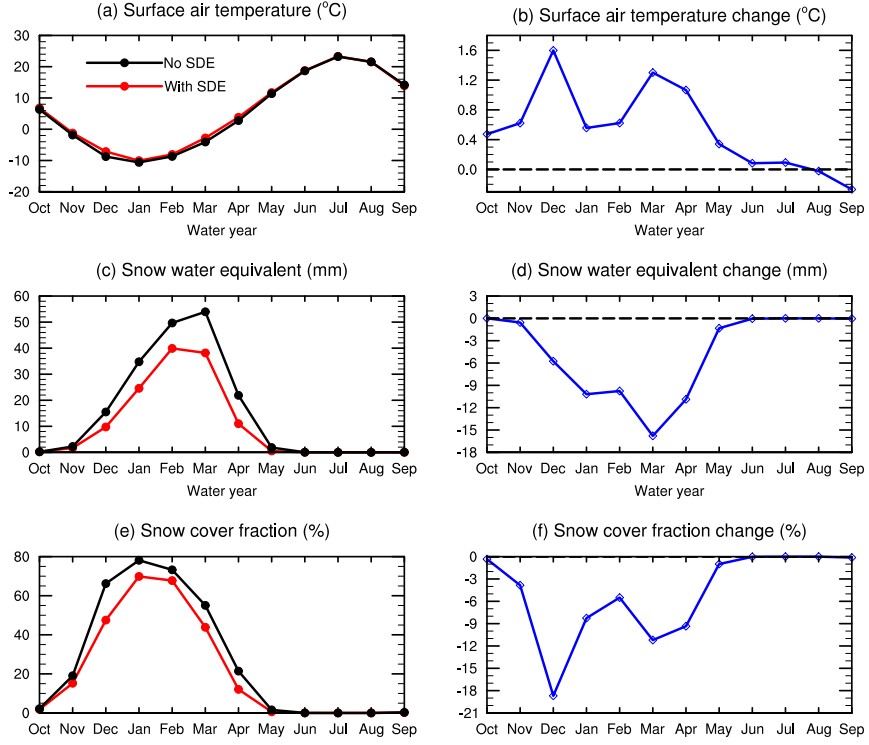


**Figure 11.** As Figure 10, but for Southwestern Wyoming.





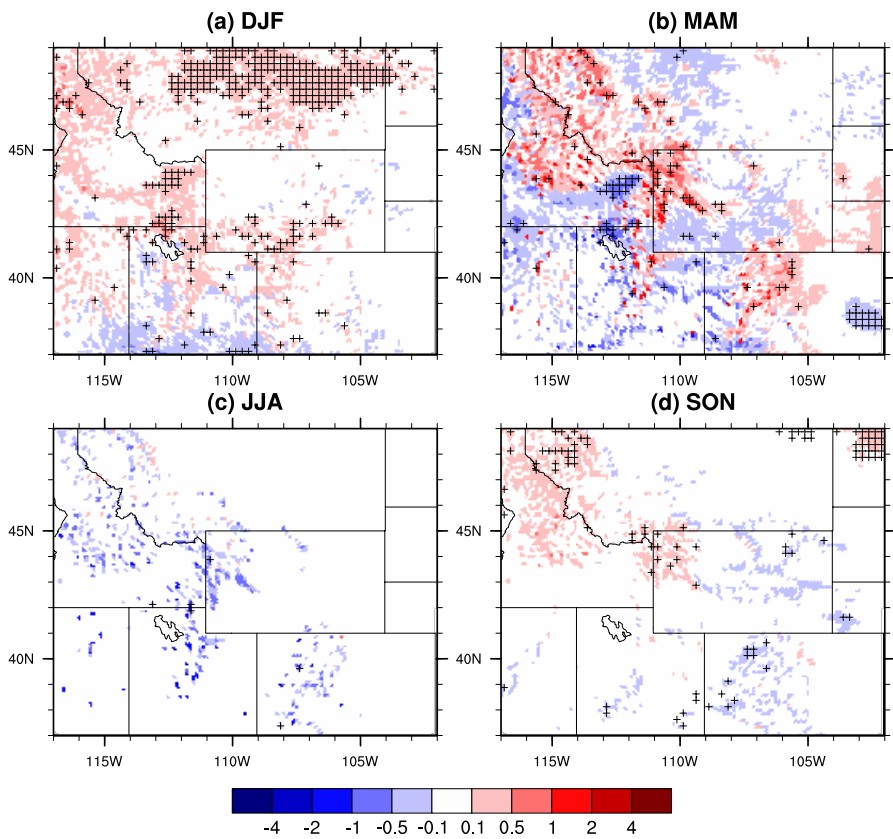


**Figure 12.** Snowmelt change (mm day$^{-1}$) due to SDE of BC and dust in four seasons:

(a) December-January-February (DJF), (b) March-April-May (MAM), (c)
June-July-August (JJA), and (d) September-October-November (SON). The crosses
denote the regions where changes induced by SDE are statistically significant at 0.1
level.





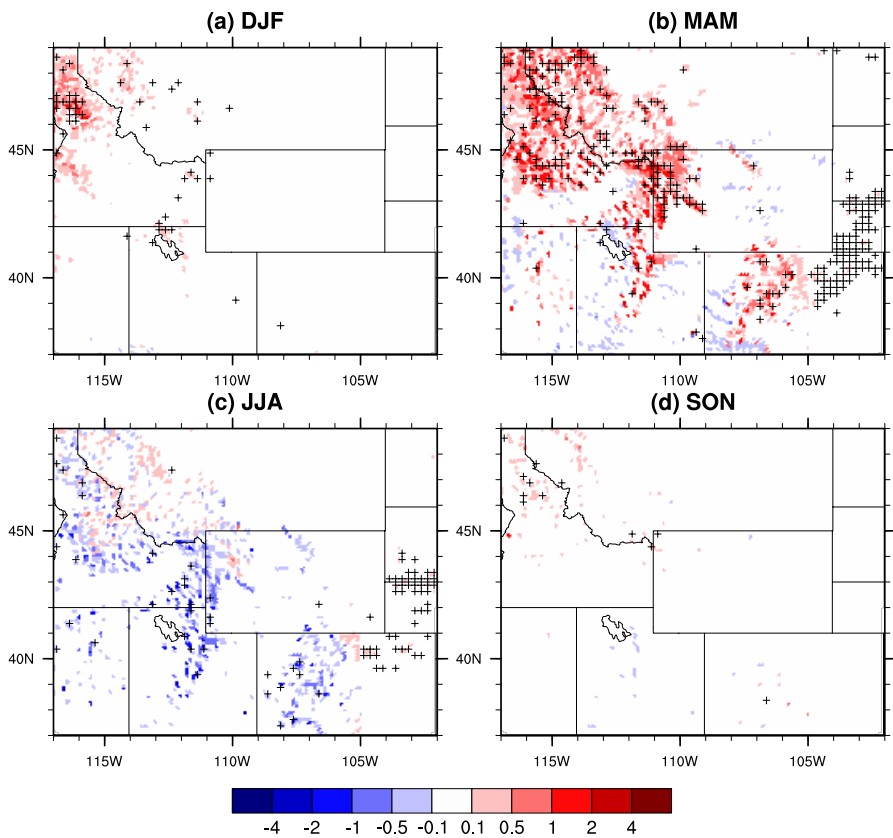


**Figure 13.** As Figure 12, but for runoff change (mm day$^{-1}$).






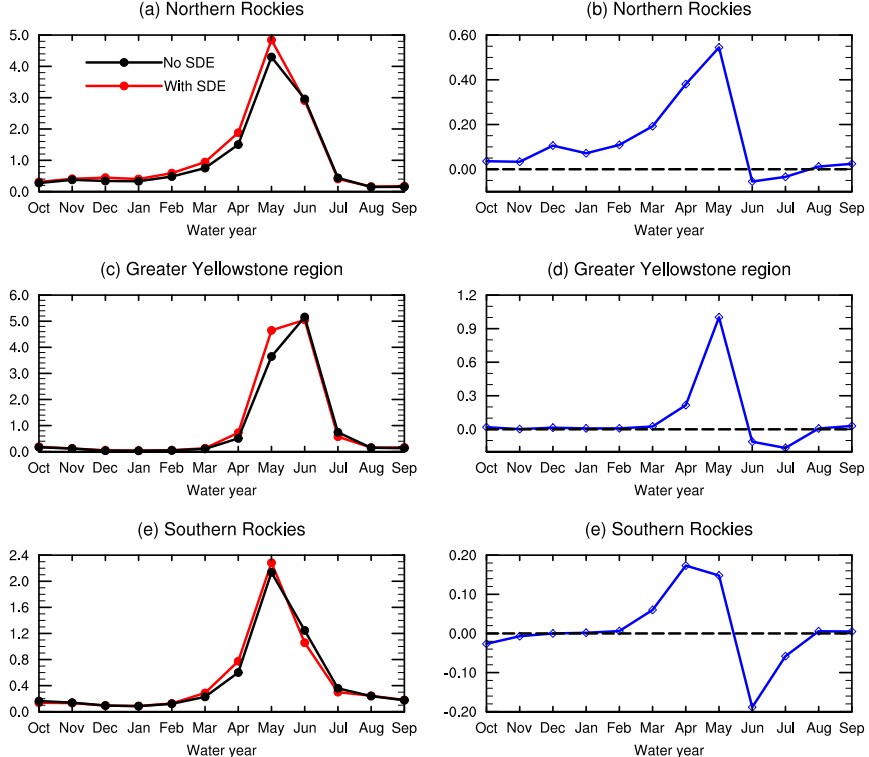


**Figure 14.** Seasonal evolution of runoff (left) and their change (right) in the Northern
Rockies (top), the Greater Yellowstone region (center), and Southern Rockies
(bottom). The unit is mm day$^{-1}$.