# Peer review of "Impacts of absorbing aerosol deposition on snowpack and hydrologic"

_Atmospheric Chemistry and Physics, 2017_

## Referee Comment (RC1) · Anonymous Referee #1 · 26 Sep 2017

This study describes a novel use of a variable resolution configuration of the Community Earth System Model (VR-CESM) to explore impacts of light absorbing aerosols in snow of the Rocky Mountains. Previously, these and other mountain ranges could not be adequately resolved in coarse resolution GCM studies to quantify impacts of aerosols on mountain snow. The configuration applied here represents the Rocky Mountains with 0.125 degree horizontal resolution, constituting a substantial improvement over previous global and even regional model simulations. This is a very thorough end-to-end study including evaluation of simulated atmospheric and in-snow aerosol
concentrations against observations followed by analysis of radiative forcings, temperature response, and hydrological response to the presence of black carbon and dust. Overall, I find this to be an excellent, logically-organized, and well-written study. I have only minor comments.

General comments:

The authors point out that simulated dust-in-snow concentrations are 1-2 orders of magnitude lower in the San Juans than measured by Skiles and Painter (2016) and simulated radiative forcing from dust is about an order of magnitude smaller. These are substantial biases. The authors also mention that dust particles larger than 10um are not included in the simulations, but comprise a majority of dust mass in measurements from this region. Is there good reason to believe that these model biases persist (and are of similar magnitude) throughout the study area, or do the authors believe these biases are somewhat unique to the San Juan Mountain area? If the former, I would consider mentioning in the abstract the omission of particles larger than 10um as a potential source of systemic bias in dust-in-snow SRE throughout the study area.

The study acknowledges that use of a coarse resolution (1.9x2.5 degrees) BC emissions inventory could have biased the simulation, which was conducted at ∼16 times higher resolution. In fact, the native resolution of the emissions inventory produced by Lamarque et al (2010) was 0.5 degrees (see abstract of that paper), so in fact finer resolution emissions could have been applied in this study. I do not suggest that the runs be conducted again, but I mention it so the authors are aware that higher resolution versions of their emissions data exist.

Please describe in more detail which version of the modal aerosol model (MAM) is applied here. i.e., is MAM3, MAM4, or MAM7 used? How, briefly, are black carbon and dust treated in this version of MAM/CESM?

Specific comments:

line 91: "except that" -> perhaps "except when" (grammar issue)

line 113: "by comparing against" -> "in comparison with"

line 132: "for for"

line 216: "all aerosols except BC (dust) as Flanner et al." -> "all aerosols except BC (i.e., only dust in this case) as in Flanner et al."

line 217: "the five regions" -> "five regions"

line 293: "snow samples at a depth of 30cm" - Here, I suspect you mean "snow samples through a depth of 30 cm" (i.e. samples collected from 0 - 30cm depth).

line 301: "is mainly contributed from" -> "consists mainly of"

line 309: "cycles" and "cycle" -> "circles" and "circle"

line 324: "although it is much weaker" -> "although they are much weaker"

line 398: "Rockiest" -> "Rockies"

line 405-408: The description of how a monthly-mean BC-in-snow concentration differs between the model and reality in cases where there is no snow during part of the month is unclear to me. Please elaborate a bit on this description, and if necessary describe any associated implications more clearly.

lines 498-502: In the discussion of dust SRE I would acknowledge again that the results do not include particles larger than 10um, which may/probably constitute the majority of dust-in-snow mass.

line 525: "This suggests that snow on the high mountains is less susceptible to the aerosol SDE" - I would re-word this. A higher snow cover fraction does not necessarily imply lower aerosol SDE. Quite often it is the opposite.

line 543: "ratio of surface air temperature change to SRE" - I suggest emphasizing that the efficacy is defined here in terms of *local* delta T and SRE.

line 559: "... which corresponds to a fraction of ..." - The meaning of this fraction (apparently a fraction of snow cover fraction) is confusing. I suggest using the terms "absolute" and "relative" to differentiate the two, or maybe just removing the relative fractions because I don't really think they are necessary.

line 565: ".., the runoff includes the surface runoff and sub-surface runoff" - Is sub-surface runoff examined at all in this study? If not, I would briefly mention that here.

Figure 3 caption: Please define "cold season"

Figure 5: There is a lot of white space in this figure. I think the axis ranges could be narrowed a bit.

Figure 8 caption: There are two references to "bottom row". The first should be "middle row"

Figure 14: Does this depict surface runoff only, or total runoff (including the sub-surface component)?

---

## Referee Comment (RC2) · Anonymous Referee #2 · 7 Oct 2017

This is a very end-to-end modeling analysis of the effects on surface air temperature and snowpack (SWE, fraction and runoff) in several regions of the western U.S. It includes a comparison of modeled near-surface atmospheric BC concentrations and mixing ratios of BC and dust in snow against observational datasets.

I have no fundamental problems with the analysis. The paper should be accepted after addressing the issues noted below. It could use some editing for English but overall is well-written, if a bit long, in part due to being repetitive in some places. I have enclosed an annotated version of the paper showing the small edits I think are needed for better

English/readability.

The following issues need addressing:

Pg 8, lines 153-154: "...CLM4 explicitly represents the snowpack (snow accumulation and melt)..." Does it also represent sublimation?

Pg. 8, lines 160-162: I think it should be explicitly pointed out that SNICAR includes the effects of feedbacks to the snowpack (grain size, melt) that are driven by albedo reduction with LAA deposition.

Pg. 9, lines 172-173 and Figure 1b: "As shown in Figure 1b, the high-resolution grids resolve well the variations of terrain in the Rocky Mountains." First: Is Figure 1.b at 0.125deg resolution? That's not clear. The figure caption just says 1b shows the terrain height within the region that is modeled at 0.125deg res – but not that the terrain height data shown in the figure is itself at 0.125deg resolution. Second: The figure just shows terrain height – there's nothing to indicate whether the terrain height at 0.125deg res "resolves well" the terrain (e.g. an actual comparison of terrain height at 0.125deg res vs at some much high res) so I'm not sure what the basis is for this assertion.

Pg. 10 lines 200-201: It is not clear here that the a-posteriori tuning factor is determined as part of this study, or if it was done as part of a previous study and you are just applying an additional adjustment factor here, based on the high-resolution model fields.

Pg. 13, line 257: Are the 80 and 94 stations "totals" all stations in existence or the total number of stations from which data are used in this analysis?

Pg. 13, lines 270-271: Two important things you need to clarify here: First, that you used the snow mixing ratios from the full snow column (not, e.g., just the surface snow mixing ratios) Second, you need to clarify how the column mixing ratio was calculated. Did you average the mass mixing ratios, or calculate the masses of BC throughout the snow column and of snow (SWE) through the whole column, then calculate the mixing ratio from that?

Pg. 15, lines 299-300: Important: The dust in snow may have a much larger size distribution than the typical tropospheric dust size distribution. Dust >10microns can be lofted from the surface but will not travel very far because they will rapidly dry-deposit to the surface (i.e. to the snow!), so they don't contribute much to the atmospheric dust but can contribute significant mass to deposited dust. For dust deposited to snow, this will of course be the case more so the closer the snow is to the dust source.

Pg. 16, lines 333-334: Important: "Overall, the model captures the magnitudes of observed near-surface BC and dust concentrations..." Here and in several other places assertions such as this are made, which give the impression that agreement is much better than in fact it is. In fact the correlation is not very good (R-squared of 0.3), and being within a factor of 5 is not necessarily representing mixing ratios well... Instead, please just state quantitatively what you found, e.g., that "the modeled concentrations are generally within a factor of 5 of the observed concentrations, and the two are moderately correlated (R-squared 0.3). Averaged across all comparison points, the model concentrations are a factor of 1.8 lower than the observed concentrations."

Pg. 18, lines 364-365: I don't think it's really shown – except in a very hand-waving way, but certainly not quantitatively – that the model "does reasonably well" in simulating the spatial variations in surface atmospheric BC. So I'd omit this sentence and let Figure 2 speak for itself, unless you want to add an analysis showing quantitatively how well spatial variations are represented.

Pg. 19, lines 384-386: "This indicates that BC and dust accumulate within the snow column...". BC and dust will be added any time snow is added, but this doesn't make the MIXING RATIO at the surface higher, so this statement is misleading. It's not clear what point you're trying to make here.

Pg. 19, lines 388-391: "As observations only sampled the snow in one day .... are given for the comparison." I don't understand what you are trying to say in this sentence; please re-write for better clarity.

Pg. 19, lines 393-394. "The model reproduces reasonably the magnitude of observed BC-in-snow mass mixing ratios at most of the stations". Again, this judgement of "reasonably" is not really justified. As with the comparison to atmospheric concentrations, please just let the data speak for itself, and give quantification of agreement (R-squared; agree within a factor of XX; mean bias. . .)

Pg. 20, lines 399-400. I don't see how this "indicates the northward transport of BC"

Pg. 20, lines 405-207: "When snow is melted completely, BC-in-snow mixing ratio will be zero, but the model will average the simulation results at different time steps to derive the mean result." I am not clear what is being said here. IMPORTANT: Does this mean the average mixing ratio includes zeros where there is no snow present? If so, this is a problem, as this will incorrectly bias the average model mixing ratios low. Modeled snow BC (or dust) mixing ratios should only be averaged across locations where snow is present. Please clarify.

Pg. 21, lines 425-427: This could also be due to compensating errors in BC deposition and snowfall.

Pg. 21, line 439: Are the TSP numbers mass concentrations or number concentrations. I'm pretty sure it must be the former, but it would be good to specify.

Pg. 23, line 461: I think it would be good to point out that this amplification in spring is due in part to feedbacks

Pg. 23, line 470: Note the correction in the annotated .pdf: SRE is a function of MIXING RATIO not MASS.

Pg. 24, lines 496-497: "For the contribution of different aerosols, BC-induced springtime SRE is larger than dust-induced SRE in the five regions." This is a repeat of sentence on pg 23, lines 464-465.

[Figure]

Pg. 25, lines 512 and 518: change "around the mountains" to "adjacent to the mountains" or "surrounding the mountains". "Around the mountains" could be misinterpreted to mean in the mountains. (Ah, the joys of English!)

Pg. 26, lines 524-525: I'd think it is SWE that's a stronger determinant here. Low snow fraction = lower area over which forcing is exerted, but with lower SWE the snow albedo feedback (via exposure of the underlying surface) occurs more readily.

Pg. 26, lines 532-533: ". . . which is likely related to the large-scale circulation change due to the aerosol SDE." Nowhere is it shown that the aerosol SDE induces a large-scale circulation change. You either need to show this here, as part of this analysis, or point to a reference where this is shown. It's not clear where this assertion is coming from.

Pg 26, line 540: "around 0.003-0.17degC". "Around 0.003" is a bit silly, since "around" implies approximate, but then you give 3 decimals of precision. Instead, say "around 0 to 0.17deg C" or (probably better) "around 0 to 0.2deg C".

Pg. 30, lines 624-626: I don't understand what you are trying to say here, regarding the aerosol SDE being "more significant" in July.

Pg. 31, line 634 "the model also reproduces observed distributions of near-surface atmospheric BC and dust. . ." vs pg 31, line 638 "BC concentrations are mostly underestimated" Which is it?? The former implies the modeled and observed values agree; the latter shows they do not.

Pg. 31, lines 641-645 (e.g. "closely related"). This is a rather optimistic qualitative statement about how the model does. As noted earlier, better is to just state quantitatively what the model vs. obs bias and correlation were.

Pg. 33, lines 674-675 "reproduces observed magnitudes" What does this mean? What is the metric here? Averaging across all sites? Please quantify.

Pg. 679: As noted earlier, the snowpack dust size distribution may skewed towards even larger sizes than the atmospheric distribution, which already has significant mass >10microns.

Figure 3: The yellow color for Utah and Nevada is pretty much invisible. Please use a different color.

Figure 5: Please state in the caption what the dashed lines represent.

Figure 8: The black crosses are really difficult to see against the dark blue. Maybe try bright yellow, at least in panels c) through f)?

Please also note the supplement to this comment:
https://www.atmos-chem-phys-discuss.net/acp-2017-799/acp-2017-799-RC2-supplement.pdf

[Figure]

**Supplement:**

[revised manuscript text omitted]

---

## Short Comment (SC1) · 19 Oct 2017

I have a minor comment about snow albedo calculations. The authors used the SNICAR model (embedded in CLM) to calculate snow albedo contaminated by aerosols. The default SNICAR setup assumes aerosol-snow external mixing and spherical snow grains. If it is true in this study, I suggest that the authors explicitly state the assumptions here. Moreover, a number of recent studies (e.g., Flanner et al., 2012; Liou et al., 2014; Dang et al., 2016; He et al., 2014, 2017) have shown that both snow grain shape and aerosol-snow internal mixing play critical roles in snow albedo calculations. Specifically, nonspherical snow grains reduces snow albedo reductions caused by light-absorbing aerosols compared with spherical snow grains, while aerosol-snow internal mixing significantly enhances snow albedo reductions relative to external mixing. It will be helpful if the authors could include these recent studies and add some discussions on this aspect and potential impacts on their current model results.

References:

Dang, C., et al. (2016), Effect of Snow Grain Shape on Snow Albedo. J. Atmos. Sci., 73, 3573–3583, doi:10.1175/JAS-D-15-0276.1

Flanner, M. G., et al. (2012), Enhanced solar energy absorption by internally-mixed black carbon in snow grains, Atmos. Chem. Phys., 12, 4699-4721, doi:10.5194/acp-12-4699-2012.

He, C., et al. (2014), Black carbon radiative forcing over the Tibetan Plateau, Geophys. Res. Lett., 41, 7806–7813, doi:10.1002/2014GL062191.

He, C., et al. (2017): Impact of Snow Grain Shape and Black Carbon-Snow Internal Mixing on Snow Optical Properties: Parameterizations for Climate Models. J. Climate, 0, doi:10.1175/JCLI-D-17-0300.1

Liou, K. N., et al. (2014), Stochastic parameterization for light absorption by internally mixed BC/dust in snow grains for application to climate models, J. Geophys. Res. Atmos., 119, doi:10.1002/2014JD021665.

---

## Author Comment (AC3) · 1 Dec 2017

We thank Dr. He for the comment and pointing out the recent relevant studies. The SNICAR model we used assumes spherical snow grains and aerosol-snow external mixing to calculate the snow albedo. We have explicitly stated the assumptions in the revised manuscript. We have also included these recent studies on the effect of snow grain shape and aerosol-snow internal mixing on snow albedo, as well as the discussions on this effect and its potential impacts on our model results: "Note that the SNICAR model we use assumes spherical snow grains and aerosol-snow external

mixing for the calculation of snowpack optical properties (Flanner et al., 2007; Oleson et al., 2010). Recent studies have shown that non-spherical snow grains play a critical role in snow albedo calculations and reduce the snow albedo reductions included by LAAs compared with spherical snow grains (e.g., Liou et al., 2014; Dang et al., 2016; He et al., 2014, 2017). Nonetheless, the knowledge of snow grain shape evolution is limited and thus spherical snow grains are assumed. Studies have also shown the significant enhancement of solar radiation absorption with larger snow albedo reductions by aerosol-snow internal mixing compared to aerosol-snow external mixing (e.g., Flanner et al., 2012; He et al., 2014; Liou et al., 2014). However, although without considering aerosol-snow internal mixing, the SNICAR model we use assumes absorption-enhancing sulfate coatings to hydrophilic BC, which can mimic BC coatings by snow and compensate the neglect of absorption-enhancement by aerosol-snow internal mixing (Flanner et al., 2007; Flanner et al., 2012). Therefore, the impacts of BC in snow shown in this study (section 4) are not necessarily biased low. Despite this, assuming dust-snow external mixing this study may underestimate the impacts of dust in snow. "

---

## Author Response (AR1)

We thank the two anonymous reviewers for their valuable comments and constructive suggestions on the manuscript. Below, we explain how the comments and suggestions are addressed and make note of the revision in the revised manuscript.

**Reviewer #1**

This study describes a novel use of a variable resolution configuration of the Community Earth System Model (VR-CESM) to explore impacts of light absorbing aerosols in snow of the Rocky Mountains. Previously, these and other mountain ranges could not be adequately resolved in coarse resolution GCM studies to quantify impacts of aerosols on mountain snow. The configuration applied here represents the Rocky Mountains with 0.125 degree horizontal resolution, constituting a substantial improvement over previous global and even regional model simulations. This is a very thorough end-to-end study including evaluation of simulated atmospheric and in-snow aerosol concentrations against observations followed by analysis of radiative forcings, temperature response, and hydrological response to the presence of black carbon and dust. Overall, I find this to be an excellent, logically-organized, and well-written study. I have only minor comments.

Reply: We thank the reviewer for his/her detailed review and encouraging comments. The text and figures are revised as the reviewer suggested.

**General comments:**

The authors point out that simulated dust-in-snow concentrations are 1-2 orders of magnitude lower in the San Juans than measured by Skiles and Painter (2016) and simulated radiative forcing from dust is about an order of magnitude smaller. These are substantial biases. The authors also mention that dust particles larger than 10um are not included in the simulations, but comprise a majority of dust mass in measurements from this region. Is there good reason to believe that these model biases persist (and are of similar magnitude) throughout the study area, or do the authors believe these biases are somewhat unique to the San Juan Mountain area? If the former, I would consider mentioning in the abstract the omission of particles larger than 10um as a potential source of systemic bias in dust-in-snow SRE throughout the study area.

Reply: We thank the reviewer for the comments. We explore more extensively these biases and associated dust size distribution. It has been recognized that dust particles with the diameter larger than 10  $\mu$ m can transport regionally for hundreds of kilometers, especially under favorable weather conditions, such as strong winds. Observations made by Reynolds et al. (2016) show that airborne dust mass concentration are mainly contributed from larger particles (diameter>10 $\mu$ m) in Utah-Colorado region. We mention that they also show large portion of dust mass in snow is from larger particles in Colorado, which we didn't mention in the previous manuscript. This supports for the importance of larger particles in snow. As the stations used in Reynolds et al. (2016) are widely distributed across the Southern Rockies, it is reasonable to believe that large portion of larger dust particles (diameter>10 $\mu$ m) exist in atmosphere and in snow across the Southern Rockies.

For the surface radiative effects (SRE), we have included the comparison of our results with another study, Skiles et al. (2015), which shows SRE by dust is smaller but in a similar magnitude (32-50 W/m2) in Grand Mesa (~150 km to the north of SBBSA) compared to that in SBBSA in San Juan Mountains (50-65 W/m2). Compared to their estimations, our simulated SRE by dust (reaching up to 2-5 W/m2) is one magnitude smaller at both stations. Therefore, these model biases of SRE may persist throughout the Southern Rockies. In the revised manuscript, we have mentioned the persistence of biases in Southern Rockies in Abstract: "Compared to previous studies based on field observations, our estimation of dust-induced SRE is generally one-order of magnitude smaller in the Southern Rockies, which is ascribed to the omission of larger particles (with the diameter >10µm) in the model. This calls for the inclusion of larger particles into the model to reduce this discrepancy".

In Greater Yellowstone region and Northern Rockies which are farther from the dust source regions (Figure 1), there is no available observation for dust size distribution in atmosphere and in snow. It is possible that dust particles with diameter>10µm may still exist, but their mass concentrations should become smaller than in Southern Rockies and the biases of SRE caused by omission of dust particles with diameter>10µm should be smaller as well. We discuss this in Section 5: "Note that such bias in SRE may become smaller in the Greater Yellowstone region and

Northern Rockies which are farther from the dust source regions than Southern Rockies.".

The study acknowledges that use of a coarse resolution (1.9x2.5 degrees) BC emissions inventory could have biased the simulation, which was conducted at  $\sim 16$ times higher resolution. In fact, the native resolution of the emissions inventory produced by Lamarque et al (2010) was 0.5 degrees (see abstract of that paper), so in fact finer resolution emissions could have been applied in this study. I do not suggest that the runs be conducted again, but I mention it so the authors are aware that higher resolution versions of their emissions data exist.

Reply: we thank the reviewer for pointing out the native resolution of the emissions inventory produced by Lamarque et al (2010). Although the native resolution is  $0.5^{\circ} \times 0.5^{\circ}$ , this dataset is further processed to be at the resolution of  $1.9^{\circ} \times 2.5^{\circ}$  for its adoption in standard CESM model (at ~1° or ~2°). It is desirable that we directly process the dataset at its native resolution ( $0.5^{\circ} \times 0.5^{\circ}$ ) for CESM model, which can benefit our high-resolution simulation to resolve more spatial variations of BC emissions. We plan to do this in the future. We have clarified in the revised manuscript in Section 2: "We note that BC emission data is natively at a resolution of  $0.5^{\circ} \times 0.5^{\circ}$  (Lamarque et al, 2010). However, it is processed to be at a relatively coarse resolution of  $1.9^{\circ} \times 2.5^{\circ}$  for adoption in standard CESM, which is used in this study." and "It is desirable to adopt BC emission at its native resolution for our high-resolution. The sensitivity of our simulation results to the resolution of BC emission will be analyzed in a separate study.".

**Please describe in more detail which version of the modal aerosol model (MAM) is applied here. i.e., is MAM3, MAM4, or MAM7 used? How, briefly, are black carbon and dust treated in this version of MAM/CESM?**

Reply: We thank the reviewer for the comment. We use MAM3 in this study. We have described MAM3 and the treatment of BC and dust within MAM3 in the revised manuscript in Section 2: "Here, we use the 3-mode version of MAM (MAM3). These three modes are aitken, accumulation, and coarse modes. In MAM3, BC is treated in the accumulation mode. BC particles are instantaneously mixed with sulfate and other components in the accumulation mode once they are emitted. Dust particles with the

diameter range of 0.1-1  $\mu$ m and 1-10  $\mu$ m are emitted into the accumulation mode and coarse mode, respectively. Airborne aerosol particles are then transported by winds and delivered back to the land surface by both dry and wet deposition, as described in Liu et al. (2012). ".

Specific comments:

*line 91: "except that" -> perhaps "except when" (grammar issue)*

Reply: Done.

line 113: "by comparing against" -> "in comparison with"

Reply: Done.

line 132: "for for"

Reply: We have deleted a redundant "for".

line 216: "all aerosols except BC (dust) as Flanner et al." -> "all aerosols except BC (i.e., only dust in this case) as in Flanner et al."

Reply: Done.

*line 217: "the five regions" -> "five regions"* Reply: Done.

line 293: "snow samples at a depth of 30cm" - Here, I suspect you mean "snow samples through a depth of 30 cm" (i.e. samples collected from 0 - 30cm depth).

Reply: We change "snow samples at a depth of 30cm" to "snow samples in the top 30 cm of the snow column".

line 301: "is mainly contributed from" -> "consists mainly of"

Reply: We change "is mainly contributed from" to "is mainly from". We think that "consists mainly of" may apply to the number of particles, but not to the mass of particles.

line 309: "cycles" and "cycle" -> "circles" and "circle"

Reply: Done.

line 324: "although it is much weaker" -> "although they are much weaker"

Reply: Done.

**line 398: "Rockiest" -> "Rockies"**

Reply: We agree with the reviewer, but this sentence is not used in the revised manuscript (we recalculate simulated BC-in-snow concentration by using daily results instead of monthly-mean results, and the analysis of comparison results is re-written.)

line 405-408: The description of how a monthly-mean BC-in-snow concentration differs between the model and reality in cases where there is no snow during part of the month is unclear to me. Please elaborate a bit on this description, and if necessary describe any associated implications more clearly.

Reply: We thank the reviewer for pointing out this. In the previous manuscript, we derive the simulated BC-in-snow concentration from monthly model result. The monthly result is an average of the results for all the timesteps during the month. If snow is not present (e.g., completely melted) in some days, BC-in-snow concentration is set to zero in these days and the monthly mean accounts for these "zero" values during the month. For observations, only the BC samples within snow (snow depth  $\geq$  5 mm or snow water equivalent  $\geq$  1 mm) are analyzed and they are not "zero".

To be consistent with the observations, in the revised manuscript, we use daily model output and derive the in-snow BC concentrations (with snow water equivalent  $\geq 1$  mm) on the same date (month/day) as the observations. By doing this, we eliminate the influence of "zero" values in the monthly model results. We clarify this in the

revised manuscript: "As our simulation period (1981-2005) does not encompass the years 2013 and 2014, we will use the daily simulation results of  $C_{BC}$  on the same month/day (or months/days; Table 1) when the observations were made (i.e., we will ignore the exact year) and compare them (means and standard deviations) with the observations. At each station, simulated daily mean  $C_{BC}$  is used only when snow is present (i.e., daily mean snow water equivalent  $\geq 1$ mm) in the simulation. 1 mm is chosen to be consistent with the minimum snow-layer thickness in observations." (Section 3).

lines 498-502: In the discussion of dust SRE I would acknowledge again that the results do not include particles larger than 10um, which may/probably constitute the majority of dust-in-snow mass.

Reply: We thank the reviewer for the comment. We add the discussion: "Note that dust-induced SRE shown here doesn't take into account dust particles larger than 10  $\mu$ m, which can constitute the majority of dust-in-snow mass (Reynolds et al., 2016). Therefore, our estimations of dust-induced SRE may be biased low."

line 525: "This suggests that snow on the high mountains is less susceptible to the aerosol SDE" - I would re-word this. A higher snow cover fraction does not necessarily imply lower aerosol SDE. Quite often it is the opposite.

Reply: We thank the reviewer for the comment. We agree with the reviewer and re-word this sentence: "Local aerosol SDE may also induce substantial impacts on the surface temperature and snowpack on the high mountains, but these impacts may be canceled out by the increase of snowfall (Figure 9f)."

line 543: "ratio of surface air temperature change to SRE" - I suggest emphasizing that the efficacy is defined here in terms of \*local\* delta T and SRE.

Reply: We change "ratio of surface air temperature change to SRE" to "ratio of local surface air temperature change to SRE over a specific region".

line 559: "... which corresponds to a fraction of ..." - The meaning of this fraction (apparently a fraction of snow cover fraction) is confusing. I suggest using the

terms "absolute" and "relative" to differentiate the two, or maybe just removing the relative fractions because I don't really think they are necessary.

Reply: We thank the reviewer for pointing out this. Following the reviewer's suggestion, we remove the relative fractions.

line 565: "..., the runoff includes the surface runoff and sub-surface runoff" - Is subsurface runoff examined at all in this study? If not, I would brielfy mention that here.

Reply: We thank the reviewer for the comment. We only show the total runoff in the previous manuscript. We combine surface runoff and subsurface runoff, as they both will flow to river and become discharge. Following the reviewer's suggestion, we clarify this and briefly mention the simulation results of surface/surface runoff in the revised manuscript: "Our simulations show that the spatial distribution and seasonal evolution of surface runoff and sub-surface runoff are generally similar to total runoff (Figure not shown), and surface runoff and subsurface runoff in the mountains. Here we only show the simulation results of total runoff, as both surface runoff and sub-surface runoff will flow to rivers and become discharge.".

**Figure 3 caption: Please define "cold season"**

Reply: We add "(winter and spring)" after "cold season".

**Figure 5: There is a lot of white space in this figure. I think the axis ranges could be narrowed a bit.**

Reply: Now we have narrowed down the horizontal and vertical axis ranges.

Figure 8 caption: There are two references to "bottom row". The first should be "middle row"

Reply: We thank the reviewer for pointing out this. The first "bottom row" is changed to "middle row".

**Figure 14: Does this depict surface runoff only, or total runoff (including the sub-surface component)?**

Reply: We also show the simulation results of total runoff in the previous manuscript. We combine surface runoff and subsurface runoff, as they both will flow to river and become discharge. We clarify in the revised manuscript by change "runoff" before "total runoff including surface and subsurface runoff".

**Reviewer #2**

This is a very end-to-end modeling analysis of the effects on surface air temperature and snowpack (SWE, fraction and runoff) in several regions of the western U.S. It includes a comparison of modeled near-surface atmospheric BC concentrations and mixing ratios of BC and dust in snow against observational datasets.

I have no fundamental problems with the analysis. The paper should be accepted after addressing the issues noted below. It could use some editing for English but overall is well-written, if a bit long, in part due to being repetitive in some places. I have enclosed an annotated version of the paper showing the small edits I think are needed for better English/readability.

Reply: We thank the reviewer for detailed review and helpful comments. The text, tables, and figures are revised as the reviewer suggested. We have also included the edits made by the reviewer.

**The following issues need addressing:**

*Pg* 8, *lines* 153-154: *"…CLM4 explicitly represents the snowpack (snow accumulation and melt)…" Does it also represent sublimation?*

Reply: Yes, CLM4 also represents sublimation. We clarify this in the revised manuscript: "...CLM4 explicitly represents the snowpack (accumulation due to snowfall and frost, loss due to sublimation, and melt)...".

**Pg.* 8, lines 160-162: I think it should be explicitly pointed out that SNICAR includes the effects of feedbacks to the snowpack (grain size, melt) that are driven by albedo reduction with LAA deposition.**

Reply: We thank the reviewer for the comment. We add the statement in the revised manuscript: "It should be mentioned that SNICAR includes the effects of feedbacks to the snowpack (grain size, melt) that are driven by snow albedo reduction due to LAA deposition."

Pg. 9, lines 172-173 and Figure 1b: "As shown in Figure 1b, the high-resolution grids resolve well the variations of terrain in the Rocky Mountains." First: Is Figure 1.b at 0.125deg resolution? That's not clear. The figure caption just says 1b shows the terrain height within the region that is modeled at 0.125deg res – but not that the terrain height data shown in the figure is itself at 0.125deg resolution. Second: The figure just shows terrain height – there's nothing to indicate whether the terrain height at 0.125deg res "resolves well" the terrain (e.g. an actual comparison of terrain height at 0.125deg res vs at some much high res) so I'm not sure what the basis is for this assertion.

Reply: We thank the reviewer for pointing out this. Figure 1b shows the terrain height used in VR-CESM, and the resolution is same as the variable resolution grid (i.e., the resolution is 0.125 degree only in the region surrounded by dashed lines and increases gradually to 1 degree outside of the region). Comparisons with United States Geological Survey (USGS) 3km data is shown in Wu et al. (2017), and the results reveal that the topography data used in VR-CESM resolves well the variations of terrain in the Rocky Mountains. We clarify this in the revised manuscript: "Figure 1b shows the spatial variations of terrain height for the variable resolution grid used in VR-CESM. Compared to United States Geological Survey (USGS) 3km topography data used in VR-CESM resolve well the variations of terrain in the Rocky Mountains (see Figure 2 of Wu et al. (2017))." (Section 2) and "(b) Terrain height (m) in the western U.S. for the variable resolution grid used in VR-CESM. The refined region at a resolution of 0.125° is surrounded by dashed lines." (Figure 1 captions).

Pg. 10 lines 200-201: It is not clear here that the a-posteriori tuning factor is determined as part of this study, or if it was done as part of a previous study and you are just applying an additional adjustment factor here, based on the high-resolution model fields.

Reply: We thank the reviewer for the comment. Before we conducted VR-CESM simulation presented in this study, we had run a test simulation using VR-CESM, which shows that surface dust concentrations are overestimated in North America. Therefore, we reduced the tuning factor (T) accordingly and conduct VR-CESM simulation for this study. We clarify in the revised manuscript: "Due to the large uncertainty in modeled dust emission, the dust emission scheme also adopts a tuning factor (*T*) to simulate the reasonable dust emission amount. Our test simulation shows that with the increase of model resolution, VR-CESM produces much higher dust concentrations compared to the observations (section 3) in North America if *T* used in the standard CESM with quasi-uniform 1° resolution is used. Therefore, for VR-CESM simulation in this study, *T* is reduced by a factor of 2.6 to produce the similar magnitudes of near-surface dust concentrations as the observations, as will be shown in section 4.1".

**Pg. 13, line 257: Are the 80 and 94 stations "totals" all stations in existence or the total number of stations from which data are used in this analysis?**

Reply: 80 and 94 are for the stations from which data are used in this analysis. We clarify this in the revised manuscript: "In total 80 and 94 stations are selected for BC and dust observations, respectively, in the western U.S. (Figure 2).".

Pg. 13, lines 270-271: Two important things you need to clarify here: First, that you used the snow mixing ratios from the full snow column (not, e.g., just the surface snow mixing ratios) Second, you need to clarify how the column mixing ratio was calculated. Did you average the mass mixing ratios, or calculate the masses of BC throughout the snow column and of snow (SWE) through the whole column, then calculate the mixing ratio from that?

Reply: We thank the reviewer for pointing out this. We clarify this in the revised manuscript: "For comparison with model simulations, we derive observed BC mass mixing ratios ( $C_{BC}$ ) in the whole snow column at sites #1-12 and #16-17 by dividing

total BC mass throughout the snow column by total snow mass throughout the snow column. At sites #13-15, the averages of  $C_{BC}$  for all the aged snow samples (from various depths and columns) were reported by Doherty et al. (2016) and are used here.".

Pg. 15, lines 299-300: Important: The dust in snow may have a much larger size distribution than the typical tropospheric dust size distribution. Dust >10microns can be lofted from the surface but will not travel very far because they will rapidly dry-deposit to the surface (i.e. to the snow!), so they don't contribute much to the atmospheric dust but can contribute significant mass to deposited dust. For dust deposited to snow, this will of course be the case more so the closer the snow is to the dust source.

Reply: We thank the reviewer for the comment. We agree with the reviewer, and will highlight the importance of large dust particles (>10 µm) in snow. We more extensively examine previous observational studies, and find that in addition to atmospheric dust particles, Reynolds et al. (2016) also measured the dust mass in snow, which show that the in-snow dust mass is mainly from dust particles with diameter > 10  $\mu$ m (consistent with the size distribution of atmospheric dust particles). This can directly support the existence of large dust particles (>10  $\mu$ m) in snow. We add the observational evidence of dust particle size in snow from Reynolds et al. (2016): "According to the observations of Reynolds et al. (2016), the mass concentration of total suspended particles (TSP) both in the atmosphere and in snow is mainly from particles with diameters larger than 10 µm in the Utah-Colorado region.". We also point out the importance of large dust particles (>10  $\mu$ m) in snow, but they are omitted in the model: "Compared to previous studies based on field observations, our estimation of dust-induced SRE is generally one-order of magnitude smaller in the Southern Rockies, which is ascribed to the omission of larger dust particles (with the diameter  $>10\mu$ m) in the model. This calls for the inclusion of larger dust particles into the model to reduce this discrepancy." (Abstract).

Pg. 16, lines 333-334: Important: "Overall, the model captures the magnitudes of observed near-surface BC and dust concentrations: : :" Here and in several other places assertions such as this are made, which give the impression that agreement is much better than in fact it is. In fact the correlation is not very good (R-squared

of 0.3), and being within a factor of 5 is not necessarily representing mixing ratios well...Instead, please just state quantitatively what you found, e.g., that "the modeled concentrations are generally within a factor of 5 of the observed concentrations, and the two are moderately correlated (R-squared 0.3). Averaged across all comparison points, the model concentrations are a factor of 1.8 lower than the observed concentrations."

Reply: We thank the reviewer for pointing out this. We have revised the analysis and state quantitatively the comparison results: "The modeled concentrations are generally within a factor of 5 of the observed concentrations, and the two are moderately correlated (the correlation coefficients (R) being 0.56 and 0.47 for BC and dust concentrations, respectively). Averaged across all comparison stations, the modeled BC concentration is a factor of 1.8 lower than the observed concentrations, and the modeled dust concentration a factor of 1.4 higher.". We also read through the manuscript, and revised the statements like this.

Pg. 18, lines 364-365: I don't think it's really shown – except in a very hand-waving way, but certainly not quantitatively – that the model "does reasonably well" in simulating the spatial variations in surface atmospheric BC. So I'd omit this sentence and let Figure 2 speak for itself, unless you want to add an analysis showing quantitatively how well spatial variations are represented.

Reply: We thank the reviewer for pointing out this. We have deleted this sentence in the revised manuscript.

Pg. 19, lines 384-386: "This indicates that BC and dust accumulate within the snow column...". BC and dust will be added any time snow is added, but this doesn't make the MIXING RATIO at the surface higher, so this statement is misleading. It's not clear what point you're trying to make here.

Reply: We thank the reviewer for pointing out this. We agree with the reviewer, and identify the reason for larger BC/dust mixing ratios in snow in spring than in winter is the larger BC/dust deposition. Therefore we delete this statement and clarify in the revised manuscript: "This is due to larger deposition of BC/dust in spring than in winter, resulting from larger northward transport of BC/dust in spring (Figure not

shown). Larger dust deposition in spring can also be partly explained by the larger dust emission in this season.".

Pg. 19, lines 388-391: "As observations only sampled the snow in one day .... Are given for the comparison." I don't understand what you are trying to say in this sentence; please re-write for better clarity.

Reply: We thank the reviewer for the comment. I use this sentence to state that the observation is based on short-term measurement (one day or tens of days), and how the model results are specifically derived for a fair comparison. In the revised manuscript, we delete this sentence as we already describe how to derive the simulation results for comparison with the observations in Section 3.

Pg. 19, lines 393-394. "The model reproduces reasonably the magnitude of observed BC-in-snow mass mixing ratios at most of the stations". Again, this judgement of "reasonably" is not really justified. As with the comparison to atmospheric concentrations, please just let the data speak for itself, and give quantification of agreement (R-squared; agree within a factor of XX; mean bias...)

Reply: We thank the reviewer for the comment. Following the reviewers' suggestion, we compare the simulated results with the observations more in the revised manuscript: "Simulated BC mixing ratios range from 8.3 ng g-1 to 30.6 ng g-1 at these sites, which are in the range of observations. Despite this, simulated BC-in-snow mass mixing ratios differ from the observations by a factor of up to 4 at some stations. Averaged across at the 17 sites, the simulated BC mass mixing ratio is 35% larger than the observed value.".

**Pg. 20, lines 399-400. I don't see how this "indicates the northward transport of BC"**

Reply: We thank the reviewer for pointing out this. We delete this statement in the revised manuscript.

*Pg. 20, lines 405-207: "When snow is melted completely, BC-in-snow mixing ratio will be zero, but the model will average the simulation results at different time steps*

to derive the mean result." I am not clear what is being said here. IMPORTANT: Does this mean the average mixing ratio includes zeros where there is no snow present? If so, this is a problem, as this will incorrectly bias the average model mixing ratios low. Modeled snow BC (or dust) mixing ratios should only be averaged across locations where snow is present. Please clarify.

Reply: We thank the reviewer for pointing out this. Yes, in the previous manuscript the monthly mean results from the model include "zero" values in some days where there is no snow present. We agree with the reviewer that this will incorrectly bias the average model mixing ratios low. To be consistent with the observations, in the revised manuscript, we use the daily BC mixing ratios when snow is present (snow water equivalent  $\geq 1$  mm) and average the simulated results on the same date (month/day) as the observations for comparison. As expected, the modeled in-snow BC mixing ratios are larger compared to the monthly mean results at the stations where snow layers are thin (e.g., sites #9, #10, #15, and #16). In the revised manuscript, we clarify that: "As our simulation period (1981-2005) does not encompass the years 2013 and 2014, we will use the daily simulation results of  $C_{BC}$ on the same month/day (or months/days; Table 1) when the observations were made (i.e., we will ignore the exact year) and compare them (means and standard deviations) with the observations. At each station, daily simulation results are used only when snow is present (i.e., daily mean snow water equivalent  $\geq 1$  mm). 1 mm is chosen to be consistent with the minimum snow-layer thickness in observations." (Section 3) and "Simulated BC mixing ratios range from 8.3 to 30.6 ng  $g^{-1}$  at these sites, which are in the range of observations. Despite this, simulated BC-in-snow mass mixing ratios differs from the observations by a factor of up to 4 at some stations. Averaged across at the 17 sites, the simulated BC mass mixing ratio is 35% larger than the observed value.." (Section 4.2).

Modeled dust mixing ratios are also averaged when snow is present: "For the simulation, we will calculate mean  $C_{dust}$  for May-June from daily  $C_{dust}$  on the days when snow is present (i.e., snow water equivalent  $\geq 10$  mm).". Snow water equivalent of 10 mm is equivalent to snow depth of 30-100 mm, which is comparable to the snow-layer interval of 3 cm in the observation.

*Pg. 21, lines 425-427: This could also be due to compensating errors in BC deposition and snowfall.*

Reply: We thank the reviewer for the comment. We add this explanation in the revised manuscript: "Another reason for the inconsistency of BC mass mixing ratios in snow and near-surface BC concentrations in the atmosphere may be related to the compensating errors in BC deposition and snowfall."

*Pg. 21, line 439: Are the TSP numbers mass concentrations or number concentrations. I'm pretty sure it must be the former, but it would be good to specify.*

Reply: They are mass concentrations. We have added "mass" before "concentrations".

*Pg. 23, line 461: I think it would be good to point out that this amplification in spring is due in part to feedbacks*

Reply: We have mentioned feedbacks in the revised manuscript: "This is because of the stronger solar insolation and larger albedo reduction due to snow aging, aerosol accumulation within snow, and feedbacks in spring."

*Pg. 23, line 470: Note the correction in the annotated .pdf: SRE is a function of MIXING RATIO not MASS.*

Reply: We have changed "mass values" to "mixing ratios".

*Pg. 24, lines 496-497: "For the contribution of different aerosols, BC-induced springtime SRE is larger than dust-induced SRE in the five regions." This is a repeat of sentence on pg 23, lines 464-465.*

Reply: We delete this sentence in the revised manuscript.

Pg. 25, lines 512 and 518: change "around the mountains" to "adjacent to the mountains" or "surrounding the mountains". "Around the mountains" could be misinterpreted to mean in the mountains. (Ah, the joys of English!)

Reply: We have changed "around the mountains" to "in the regions adjacent to the mountains".

*Pg. 26, lines 524-525: I'd think it is SWE that's a stronger determinant here. Low snow fraction = lower area over which forcing is exerted, but with lower SWE the snow albedo feedback (via exposure of the underlying surface) occurs more readily.*

Reply: We thank the reviewer for the comment. We agree with the reviewer and have changed this sentence to "For example, winter and spring snow water equivalent is mostly above 50 mm on the high mountains (see Figure 8 of Wu et al. (2017)).".

Pg. 26, lines 532-533: "... which is likely related to the large-scale circulation change due to the aerosol SDE." Nowhere is it shown that the aerosol SDE induces a largescale circulation change. You either need to show this here, as part of this analysis, or point to a reference where this is shown. It's not clear where this assertion is coming from.

Reply: We thank the reviewer for the comment. This assertion is based on our simulation results and we apology for not showing the results in the previous manuscript. In the revised manuscript, we have added the analysis of simulation results and clarified the assertion: "The increase of snowfall in Figure 9 is likely related to the large-scale circulation change due to aerosol SDE. Figure 10 shows wintertime tropospheric temperature and zonal winds in CTL and NoSDE simulations and their difference. In the NoSDE simulation, we have turned off the SDE not only in the Rocky Mountain region, but also in other regions of the globe. Due to aerosol SDE, temperature is increased in the high-latitudes of northern hemisphere (Figure 10c), which can reduce the meridional temperature gradient, thus leading to the weakening polar jet stream north of 50 °N (Figure 10f). This suggests a shift to a more meridional wind pattern in winter, which can enhance the broader meanders and thus the formation of winter storms (Wu et al., 2017). Enhanced winter storm activity further reduces surface temperature to the north of Rocky Mountain region as well as in the northern part of Rocky Mountain region (Figures 8a and 10c). This, together with the increased temperature in the southwestern U.S. and southern part of Rocky Mountain region, increases meridional temperature gradient and leads to stronger westerly at 30-45°N (Figure 10f). Stronger westerly at 30-45 °N favors the water vapor transport from the Pacific Ocean. The enhance of winter storm activity and water vapor transport may lead to the increase of precipitation (mainly in terms of snowfall in winter). In spring, the change in temperature and zonal winds is similar to that in winter, but with a northward shift of the patterns as a result of northward movements of the polar jet stream and westerlies in spring (Figure not shown). Therefore, the change of snowfall is likely a result of circulation change induced by SDE from both the Rocky Mountain region and remote regions. It is worth isolating the impacts of SDE from the Rocky Mountain region and remote regions (e.g., high-latitudes) in the future."

*Pg* 26, line 540: "around 0.003-0.17degC". "Around 0.003" is a bit silly, since "around" implies approximate, but then you give 3 decimals of precision. Instead, say "around 0 to 0.17deg C" or (probably better) "around 0 to 0.2deg C".

Reply: We have changed "around 0.003-0.17 °C" to "around 0-0.2 °C".

**Pg. 30, lines 624-626: I don't understand what you are trying to say here, regarding the aerosol SDE being "more significant" in July.**

Reply: We stated aerosol SDE in July is more significant in terms of the larger relative change of runoff (the ratio of absolute runoff change to original runoff). To clarify, we have deleted "more significant" in the revised manuscript: "Runoff is relatively smaller in July versus in previous months, and aerosol SDE can reduce the runoff by 0.04 (8%), 0.17 (23%), and 0.06 mm day-1 (16%) in the three regions, respectively".

Pg. 31, line 634 "the model also reproduces observed distributions of near-surface atmospheric BC and dust..." vs pg 31, line 638 "BC concentrations are mostly underestimated" Which is it?? The former implies the modeled and observed values agree; the latter shows they do not.

Reply: We thank the reviewer for pointing out this. The former indicates the general spatial patterns are similar for simulation and observations, such as larger BC concentrations in West Coast/Southwester U.S. and smaller BC concentrations in

Rocky Mountains/Great Plain. The latter applies to the comparison of BC concentrations in the Rocky Mountains. To clarify, we have deleted the former statement and only emphasized the comparison in the Rocky Mountain region: "Here we show that the model simulates similar magnitude of near-surface dust concentrations at most stations in the Rocky Mountain region compared to IMPROVE observations. The model tends to underestimate near-surface atmospheric BC concentrations mostly by a factor of 1.5-5 in the Rocky Mountain region.".

**Pg. 31, lines 641-645 (e.g. "closely related"). This is a rather optimistic qualitative statement about how the model does. As noted earlier, better is to just state quantitatively what the model vs. obs bias and correlation were.**

Reply: We thank the reviewer for the comment. In the revised manuscript, we have deleted "Simulated aerosol-in-snow concentrations are closely related to the distributions of both snowpack and near-surface atmospheric aerosol concentrations." and state quantitatively the comparison result: "Simulated BC-in-snow concentrations ranges from 2 to 50 ng g-1 in the Rocky Mountain region, and they are 35% larger than the observations for the average at the 17 sites.".

**Pg. 33, lines 674-675 "reproduces observed magnitudes" What does this mean? What is the metric here? Averaging across all sites? Please quantify.**

Reply: We thank the reviewer for the comment. We mean the magnitudes of simulated BC-in-snow concentrations are comparable to at most stations (i.e., in the range of observations). To clarify, we have quantify the comparison result: "however, overestimates BC-in-snow concentrations by 35% for the average across the 17 observational sites.".

Pg. 679: As noted earlier, the snowpack dust size distribution may skewed towards even larger sizes than the atmospheric distribution, which already has significant mass>10microns.

Reply: We thank the reviewer for the comment. In the revised manuscript, we have added the observation evidence from Reynolds et al. (2016) for significant

contribution of larger particles to total dust mass. We have also emphasized the importance of larger particles in Abstract.

**Figure 3: The yellow color for Utah and Nevada is pretty much invisible. Please use a different color.**

Reply: We have changed the yellow color to black color in the revised manuscript.

**Figure 5: Please state in the caption what the dashed lines represent.**

Reply: We have added in the caption "The 1:1 (solid) and 1:5/5:1 (dash) lines are plotted for reference."

**Figure 8: The black crosses are really difficult to see against the dark blue. Maybe try bright yellow, at least in panels c) through f)?**

Reply: We thank the reviewer for the comment. We try bright yellow, but it looks a little messy. Therefore, we change the color scheme by not using the dark blue and keep the black crosses. The revised Figure 8 looks more clear now.

Please also note the supplement to this comment: https://www.atmos-chem-phys-discuss.net/acp-2017-799/acp-2017-799-RC2-supple ment.pdf

Reply: We thank the reviewer for the edits, which improves the manuscript. We have includes these edits in the revised manuscript.

[revised manuscript text omitted]

**Chenglai 11/12/17 9:52 PM Deleted: either Chenglai 11/12/17 9:52 PM Deleted: or Chenglai 11/12/17 9:53 PM Deleted: the**

Chenglai 11/12/17 9:53 PM Deleted: For Chenglai 11/12/17 9:53 PM Deleted: , they Chenglai Wu 11/27/17 2:56 PM Deleted: that Chenglai Wu 11/27/17 10:00 AM Deleted: ; Oaida et al., 2015

| 122 | constraints of | f computational | l resources ( | (e.g., | Haarsma et | al., 201 | l 6). | Instead, | using |
|-----|----------------|-----------------|---------------|--------|------------|----------|-------|----------|-------|
|-----|----------------|-----------------|---------------|--------|------------|----------|-------|----------|-------|

123 VR-GCMs is a more economic approach and has gained increasing utility in recent

124 years (e.g., Zarzycki et al, 2014a, b; Sakaguchi et al., 2015).

| 125 | A variable-resolution | version of the | Community | Earth Sy | stem Model |
|-----|-----------------------|----------------|-----------|----------|------------|
|-----|-----------------------|----------------|-----------|----------|------------|

- 126 (VR-CESM) has been developed (Zarzycki et al., 2014a, b). With a refined high
- 127 resolution, VR-CESM has shown significant improvements of the Atlantic tropical
- 128 storms (Zarzycki and Jablonowski, 2014) and South America orographic precipitation

129 (Zarzycki et al., 2015). The model has also been used in the regional climate

130 simulations over the western U.S., and results show the VR-CESM is capable of

- 131 reproducing the spatial patterns and the seasonal evolution of temperature,
- 132 precipitation, and snowpack in the Sierra Nevada (Huang et al., 2016; Rhoades et al.,
- 133 2016) and Rocky Mountains (Wu et al., 2017). In particular, VR-CESM simulates
- 134 reasonably the magnitude of snow water equivalent, the timing of snow water
- equivalent peaks, and the duration of snow cover in the Rocky Mountains, as shown
- 136 in comparison, with Snow Telemetry (SNOTEL) and MODIS (Moderate Resolution
- 137 Imaging Spectroradiometer) snow cover observations (Wu et al., 2017).
- 138 Following the evaluation study of Wu et al. (2017), here we use VR-CESM to
- 139 investigate the impacts of LAAs in snow (BC and dust) on the snowpack and
- 140 hydrologic cycles over the Rocky Mountains. By comparing the two VR-CESM
- simulations with and without LAAs in snow, we examine the impacts on surface
- 142 radiative transfer, temperature, snowpack, and runoff induced by LAAs in snow. To

| Chenglai 11/12/17 9:56 PM |
|---------------------------|
| Deleted: in               |
| Chenglai 11/12/17 9:58 PM |
| Deleted: that             |

**Chenglai 11/12/17 9:58 PM Deleted: in**

[revised manuscript text omitted]

Chenglai Wu 11/27/17 10:15 AM Deleted: . Chenglai 11/29/17 12:52 AM Deleted: BC mixing ratios in snow

Chenglai 11/12/17 10:09 PM Deleted: at Chenglai 11/12/17 10:09 PM Deleted: at

- 373 calibration is applied to the dataset of Doherty et al. (2014) in our study, and thus the
- 374 observations of Doherty et al. (2014) and Doherty et al. (2016) used here are
- 375 comparable

| 376 | In addition, Skiles and Painter (2016b) made daily measurements of                          |
|-----|---------------------------------------------------------------------------------------------|
| 377 | BC-in-snow with an SP2 in the Senator Beck Basin Study Area (SBBSA) in the San              |
| 378 | Juan Mountains during a period of two months (late March to middle May) in 2013.            |
| 379 | The locations and sample dates as well as the measurements for these stations are           |
| 380 | given in Table 1. For comparison with model simulations, we derive observed BC              |
| 381 | mass mixing ratios ( $C_{BC}$ ) in the whole snow column at sites #1-12 and #16-17 by       |
| 382 | dividing total BC mass throughout the snow column by total snow mass throughout             |
| 383 | the snow column. At sites #13-15, the averages of $C_{BC}$ for all the aged snow samples    |
| 384 | (from various depths and columns) were reported by Doherty et al. (2016) and are            |
| 385 | used here. If measurements of $C_{BC}$ on multiple days were made, the means and     |
| 386 | standard deviations of $C_{BC}$ are given. As our simulation period (1981-2005) does not    |
| 387 | encompass the years 2013 and 2014, we will use the daily simulation results of $C_{BC_{-}}$ |
| 388 | on the same month/day (or months/days; Table 1) when the observations were made             |
| 389 | (i.e., we will ignore the exact year) and compare them (means and standard deviations       |
| 390 | with the observations, At each station, simulated daily simulation results are used         |
| 391 | only when snow is present (i.e., daily mean snow water equivalent $\geq 1$ mm), 1 mm is     |
| 392 | chosen to be consistent with the minimum snow-layer thickness in observations,              |
|     |                                                                                             |

Chenglai 11/12/17 10:10 PM

| Deleted: compare                              |
|-----------------------------------------------|
| Chenglai Wu 11/27/17 10:18 AM                 |
| Deleted: monthly mean                         |
| Chenglai Wu 11/27/17 10:17 AM                 |
| Deleted: for 25 years (1981-2005) by focusing |
| more on the general magnitudes and spatial    |
| distributions of $C_{BC}$                     |
| Chenglai 11/29/17 10:11 PM                    |
| Deleted: mean $C_{BC}$ is derived      |
| Chenglai 11/29/17 10:12 PM                    |
| Deleted: in the simulation             |
| Chenglai Wu 11/27/17 10:18 AM                 |
| Deleted: the mean results for the month (or   |
| months) when the observations were made, as   |
| well as the maximum and minimum $C_{BC},are$  |
| derived from the 25-year simulation and       |
| compared to the observations                  |

| 406 | There are few observations of dust mass mixing ratio in snow ( $C_{dust}$ ) in the                               |  |
|-----|------------------------------------------------------------------------------------------------------------------|--|
| 407 | Rocky Mountain region. To our best knowledge, the only published observations                                    |  |
| 408 | were conducted at two sites in Southern Rockies: one in the Senator Beck Basin Study                             |  |
| 409 | Area (SBBSA) in the San Juan Mountains with at least 9-year (2005-2013) records                                  |  |
| 410 | (Painter et al., 2012; Skiles et al., 2015; Skiles and Painter, 2016a, 2016b); the other                         |  |
| 411 | in the Grand Mesa (~150 km to the north of SBBSA) with at least 4-year (2010-2013)                               |  |
| 412 | records (Skiles et al., 2015). Snow samples in the top 30 cm of the snow column were                             |  |
| 413 | collected at irregular time intervals from March to June. Here we will use the                                   |  |
| 414 | end-of-year (EOY) C dust , which was reported from the samples collected just prior to                |  |
| 415 | snow depletion and consisted of the majority of dust in the snow column (Skiles et al.,                          |  |
| 416 | 2015; Skiles and Painter, 2016a). For the simulation, we will calculate mean $C_{dust}$ for                      |  |
| 417 | May-June from daily C dust on the days when snow is present (i.e., snow water                         |  |
| 418 | equivalent $\geq 10 \text{ mm}$ , Another consideration is that observed C dust contains all the dust |  |
| 419 | particles while simulated C dust only accounts for the dust particles with diameters                  |  |
| 420 | smaller than 10 µm. According to the observations by Reynolds et al. (2016), the                                 |  |
| 421 | mass concentration of total suspended particles (TSP) both in the atmosphere and in                              |  |
| 422 | snow is mainly from particles with diameters larger than 10 $\mu$ m in the Utah-Colorado                         |  |
| 423 | region. This will affect the model comparison with the observations, which will be                               |  |
| 424 | discussed in section 4.                                                                                          |  |
|     |                                                                                                                  |  |

- 425 **4. Results**
- 426 4.1 Spatial patterns of near-surface aerosol concentrations

17

Chenglai 11/12/17 10:15 PM

Chenglai Wu 11/30/17 2:53 PM Deleted: (in Southern Rockies) Chenglai Wu 11/30/17 4:30 PM Deleted: 2007,

Chenglai Wu 11/27/17 3:39 PM Deleted: at a depth of Chenglai Wu 11/27/17 10:22 AM Deleted: for each dust event Chenglai Wu 11/27/17 10:22 AM Deleted: May

Chenglai 11/29/17 10:16 PM Deleted: results Chenglai Wu 11/27/17 10:22 AM Deleted: We will compare simulated Cdust for the whole snow columns with the observations (for 0-30 cm depth), but acknowledge that Cdust may vary in the snow underneath 0-30 cm snow layer. Chenglai Wu 12/1/17 10:31 AM Deleted: of Chenglai Wu 11/27/17 4:47 PM Deleted: contributed
| 441 | Before we examine the impacts of aerosol deposition onto snow, we will first                      |  |
|-----|---------------------------------------------------------------------------------------------------|--|
| 442 | evaluate the aerosol simulations by the model. Figure 2 shows the spatial patterns of             |  |
| 443 | cold season (winter and spring) mean emission fluxes and near-surface concentrations              |  |
| 444 | of BC and dust in the western U.S. from the VR-CESM simulation. The IMPROVE                       |  |
| 445 | stations are also denoted by circles with larger circle sizes indicating higher observed          |  |
| 446 | near-surface BC/dust concentrations. In the model, the BC emission flux is prescribed             |  |
| 447 | and is largest in the Pacific Coast and southern Arizona. BC emission fluxes are                  |  |
| 448 | relatively large in central-northern Colorado and Northwestern Utah, where large                  |  |
| 449 | metropolises are located. Corresponding to the patterns of BC emission flux,                      |  |
| 450 | simulated near-surface BC concentrations (>100 ng m -3 ) are also higher in these      |  |
| 451 | regions. A band with relatively high near-surface BC concentrations around 50-100                 |  |
| 452 | ng m -3 is also found in southern Idaho, to the west of the Greater Yellowstone region |  |
| 453 | and to the south of Northern Rockies, indicating the transportation of BC around the              |  |
| 454 | mountains. Near-surface BC concentrations decrease at higher elevations. The spatial              |  |
| 455 | patterns simulated by the model are generally consistent with observations, e.g.,                 |  |
| 456 | higher BC concentrations in the source regions and lower in the mountains.                        |  |
| 457 | Dust sources are located in the dry regions with exposed bare soils, such as the                  |  |
| 458 | southwestern U.S. (southern California, western Arizona, and southern New Mexico),                |  |
| 459 | the northern Mexico, the Great Basin, and the Colorado Plateau. Dust emissions are                |  |
| 460 | also found in the Great Plains, although they are much weaker. In the Great Plains                |  |
| 461 | agricultural activities can disturb the soil, making it vulnerable to wind erosion                |  |

Chenglai Wu 11/27/17 5:22 PM Deleted: it is

Chenglai Wu 11/27/17 5:21 PM Deleted: y

Chenglai Wu 11/27/17 5:21 PM

| 465 | (Ginoux et al., 2012). Simulated cold season mean dust concentrations are higher                                     |                                                                               |
|-----|----------------------------------------------------------------------------------------------------------------------|-------------------------------------------------------------------------------|
| 466 | (10-500 $\mu g~m^{\text{-3}})$ in the source regions, but decrease dramatically (0.1-5 $\mu g~m^{\text{-3}})$ to the |                                                                               |
| 467 | mountains. Compared to the observations, the model reproduces the spatial patterns of                                |                                                                               |
| 468 | near-surface dust concentrations with higher concentrations in the southwestern part                                 |                                                                               |
| 469 | of US. However, the model tends to overestimate the dust concentrations in Utah,                                     |                                                                               |
| 470 | indicating that dust emission may be overestimated there.                                                            |                                                                               |
| 471 | Comparisons of modeled and observed near-surface BC/dust concentrations at                                           |                                                                               |
| 472 | the IMPROVE stations are further shown in Figure 3. The modeled concentrations are                                   | Chenglai Wu 11/29/17 11:32 AM                                                 |
|     |                                                                                                                      | Deleted: Overall, t                                                           |
| 473 | generally within a factor of 5 of the observed concentrations, and the two are                                       | Chenglai Wu 11/29/17 11:32 AM                                                 |
| 474 | moderately correlated (the correlation coefficients (R) being 0.56 and 0.47 for BC and                               | Deleted: captures the magnitudes of near-surface BC and dust           |
| 475 | dust concentrations, respectively). Averaged across all comparison stations, the                                     | Chenglai Wu 11/29/17 11:33 AM Deleted: with the differences betwee     |
|     |                                                                                                                      | observations and simulations                                                  |
| 476 | modeled BC concentration is a factor of 1.8 lower than the observed concentrations,                                  | Chenglai Wu 11/29/17 11:33 AM                                                 |
| 477 | and the modeled dust concentration a factor of 1.4 higher. The model tends to                                        | Deleted: for most of the stations                                      |
| 477 | and the modeled dust concentration a factor of 1.4 mgher. The model tends to                                         | Deleted: The model simulates simila                                           |
| 478 | systematically underestimate observed near-surface BC concentrations in                                              | magnitudes of observed near-surface                                           |
| 479 | Utah-Nevada regions, the Rocky Mountains, and the Great Plains, where the stations                                   | concentrations at the stations along the Coast and in the Southwestern U.S. H |
| 400 | and based at the second maximum (Figure 2t). In marticular shares d                                                  |                                                                               |
| 480 | are located downwind of source regions (Figure 2b). In particular, observed                                          |
Chenglai 11/12/17 11:48 PM                                                |
| 481 | near-surface BC concentrations are underestimated mostly by a factor of 1.5-5 in the                                 | Chenglai Wu 11/30/17 10:10 AM                                                 |
|     |                                                                                                                      | Deleted: about                                                                |
| 482 | Rocky Mountains. The underestimation of near-surface BC concentrations in these                                      | Chenglai Wu 11/29/17 12:01 PM                                                 |
| 483 | regions may suggest that transport of BC in our simulations is too weak. This                                        | Deleted: two                                                                  |
| 484 | deficiency may also be ascribed to local BC sources (e.g. Doherty et al. 2014) not                                   | Chenglai 11/12/17 10:20 PM                                                    |
|     |                                                                                                                      | Deleted: the                                                                  |
| 485 | resolved by the prescribed BC emission in the model (e.g. at $1.9 \times 2.5^{\circ}$ resolution). For               |                                                                               |

eleted: Overall, t nenglai Wu 11/29/17 11:32 AM eleted: captures the magnitudes of observed ar-surface BC and dust nenglai Wu 11/29/17 11:33 AM eleted: with the differences between servations and simulations nenglai Wu 11/29/17 11:33 AM eleted: for most of the stations nenglai Wu 11/29/17 12:00 PM eleted: The model simulates similar agnitudes of observed near-surface BC ncentrations at the stations along the West bast and in the Southwestern U.S. However, t

**nenglai 11/12/17 11:48 PM eleted: in the nenglai Wu 11/30/17 10:10 AM eleted: about nenglai Wu 11/29/17 12:01 PM eleted: two**

| 500 | dust, although the model overestimates near-surface dust concentrations for most of                             |  |
|-----|-----------------------------------------------------------------------------------------------------------------|--|
| 501 | the stations near the dust sources (southwestern U.S., Utah, and Nevada), the model                             |  |
| 502 | simulates reasonably the magnitude of near-surface dust concentrations in the Rocky                      |  |
| 503 | Mountains. This may also be associated with underestimated transport in the model,                              |  |
| 504 | consistent with the low bias in near-surface BC concentrations in downwind regions.                      |  |
| 505 | Note that although only the BC and dust emission fluxes over the western U.S.                                   |  |
| 506 | are shown in Figure 2, long-range transport of these aerosols from other regions (e.g.,                         |  |
| 507 | Asia and Africa) can also contribute to BC (e.g., Zhang et al., 2015) and dust (Wells                           |  |
| 508 | et al., 2007) concentrations in the western U.S. In addition , there are substantial                     |  |
| 509 | variations of aerosol emission in the western U.S. As mentioned in section 2, although                          |  |
| 510 | we adopt VR-CESM with a refined high resolution $(0.125^\circ)$ in the Rocky Mountains,                         |  |
| 511 | we use a coarse resolution gridded emission dataset (i.e., $1.9^{\circ} \times 2.5^{\circ}$ ) for BC. For dust, |  |
| 512 | the small-scale variations of dust emissions can be represented in the model as it is                           |  |
| 513 | calculated online in the model. However, dust emission depends on many variables                                |  |
| 514 | such as near-surface winds, soil moisture, vegetation cover, and soil texture (Oleson                           |  |
| 515 | et al., 2010; Wu et al., 2016), which may themselves be biased. In particular, in Utah                          |  |
| 516 | and Nevada, simulated near-surface dust concentrations are about 2-3 times as large                             |  |
| 517 | as observed , indicating significant overestimation of dust emissions in the region,              |  |
| 518 | 4.2 Aerosol-in-snow concentrations                                                                              |  |
| 519 | Figure 4 shows the spatial distributions of BC and dust mass mixing ratios in                                   |  |
| 520 | snow in winter and spring from VR-CESM simulations. BC-in-snow mass mixing                                      |  |

Chenglai Wu 11/29/17 12:28 PM Deleted: well

Chenglai 11/12/17 10:21 PM Deleted: as indicated in the bias of Chenglai 11/12/17 10:21 PM Deleted: the

Chenglai 11/12/17 10:22 PM Deleted: Meanwhile Chenglai 11/12/17 11:47 PM Deleted: also

| Chenglai 11/12/17 11:45 PM                         |
|----------------------------------------------------|
| Deleted: observations                              |
| Chenglai 11/12/17 11:45 PM                         |
| Deleted: the s                                     |
| Chenglai Wu 11/29/17 12:30 PM                      |

[revised manuscript text omitted]
 willing Chenglai Wu 11/27/17 10:31 AM Deleted: Another reason may be that observations show Chenglai Wu 11/27/17 10:32 AM Deleted: Chenglai Wu 11/27/17 10:32 AM Deleted: at site #16 and Chenglai Wu 11/27/17 10:33 AM Deleted:

Chenglai Wu 11/27/17 10:34 AM Deleted: Therefore, VR-CESM may also underestimate the BC-in-snow mass mixi....[2]

| 653 | inconsistency of BC mass mixing ratios in snow and near-surface BC concentrations                              |
|-----|----------------------------------------------------------------------------------------------------------------|
| 654 | in the atmosphere may be related to the compensating errors in BC deposition and                               |
| 655 | snowfall. This inconsistency may also be related to the snow aging/melting and                                 |
| 656 | BC-in-snow accumulation and flushing-out, which are associated with large                                      |
| 657 | uncertainties (Flanner et al., 2007; Qian et al., 2014).                                                       |
| 658 | For dust in snow, the simulated mean dust mass mixing ratio in snow in                                         |
| 659 | May-June is 31.0 (27.8), µg g -1 in the San Juan Mountains (Grand Mesa), with the                   |
| 660 | standard deviation, minimum, and maximum being 20.4 (10.0) $\mu$ g g -1 , 8.9 (14.8) $\mu$ g        |
| 661 | $g^{-1}$ , and 81.4 (50.4) $\mu g g^{-1}$ , respectively. These, values are one to two orders of               |
| 662 | magnitude smaller compared to the observed mixing ratios from Skiles et al. (2015),                     |
| 663 | which showed that, at the end of the snow season, the total dust-in-snow mass mixing                           |
| 664 | ratios range from 0.2 to 4.8 mg g -1 and from 0.6-1.7 mg g -1 , respectively, in the San |
| 665 | Juan Mountains and Grand Mesa. Much smaller dust-in-snow mass mixing ratios in                                 |
| 666 | the simulations may be ascribed to the fact that the model only accounts for dust                              |
| 667 | particles with diameters smaller than 10 $\mu m,$ while the observations include all the                       |
| 668 | sizes of dust particles in the snow. Observation by Reynolds et al. (2016) in the                       |
| 669 | Colorado region showed that mass concentrations of dust particles in snow are mostly                           |
| 670 | from larger particles with diameters larger than 10 $\mu$ m. Therefore, the model may                          |
| 671 | underestimate the impacts of dust deposition into snow, Dust impacts calculated in                             |
| 672 | this study, which will be discussed below, should be regarded as those from the dust                           |
| 673 | particles with diameters smaller than 10 µm.                                                                   |

Chenglai Wu 11/27/17 10:34 AM Deleted: spring Chenglai Wu 11/27/17 10:35 AM Deleted: 19.6 Chenglai Wu 11/27/17 10:35 AM Deleted: . The simulated Chenglai Wu 11/27/17 10:36 AM  $\ensuremath{\textbf{Deleted:}}$  of dust mass mixing ratios for 1981-2005 are 22.4  $\mu g~g^{\text{-1}},~5.3~\mu g~g^{\text{-1}},$  and 118.7 μg g-1 Chenglai Wu 11/27/17 10:36 AM Deleted: is Chenglai Wu 11/27/17 10:36 AM Deleted: is Chenglai 11/12/17 10:27 PM Deleted: observations of Chenglai Wu 11/27/17 10:36 AM Deleted: and Painter Chenglai Wu 11/27/17 10:36 AM Deleted: 6a Chenglai 11/12/17 11:42 PM Deleted: Actually, an observation made Chenglai Wu 11/27/17 10:38 AM Deleted: Utah-Chenglai Wu 11/27/17 10:38 AM **Deleted:** total suspended particles (TSP) Chenglai 11/29/17 10:35 PM Deleted: the atmosphere Chenglai 11/12/17 10:30 PM Deleted: mainly contributed Chenglai 11/12/17 10:30 PM Deleted: , and Chenglai 11/12/17 10:30 PM Deleted: the Chenglai 11/12/17 10:30 PM Deleted: d Chenglai 11/12/17 10:30 PM Deleted: can

**694 4.3 Surface radiative effect (SRE) by aerosol-in-snow**

| 695 | Figure 6 shows the spatial distribution of instantaneous surface radiative effect          |                       |
|-----|--------------------------------------------------------------------------------------------|-----------------------|
| 696 | (SRE) due to BC- and dust-induced snow albedo change, respectively, in winter              |                       |
| 697 | (December-January-February) and spring (March-April-May). Due to the decrease of           |                       |
| 698 | surface albedo, surface net shortwave radiation is increased. The spatial patterns of      |                       |
| 699 | SRE are determined by both the amount of aerosol in snow and the snowpack_                 |                       |
| 700 | distribution (snow depth and snow cover fraction). Finer-scale structures of SRE in        | $\left \right\rangle$ |
| 701 | the Rocky Mountains and the adjacent regions are simulated by VR-CESM with a               |                       |
| 702 | higher horizontal resolution compared to previous simulations by coarse-resolution         |                       |
| 703 | GCMs (e.g., Flanner et al., 2009; Yasunari et al., 2015). The SRE is generally above       |                       |
| 704 | $0.2 \text{ W m}^{-2}$ over the mountains especially in the Greater Yellowstone region and |                       |
| 705 | Southern Rockies. SRE can reach similar magnitudes on the southern periphery of            |                       |
| 706 | Northern Rockies and west side of the Greater Yellowstone region, where higher             |                       |
| 707 | near-surface atmospheric BC/dust concentrations and BC/dust-in-snow mass mixing            |                       |
| 708 | ratios are simulated (Figures 2 and 4). SRE is stronger in spring than in winter for       |                       |
| 709 | both BC and dust, which is consistent with previous studies (Flanner et al., 2009;         |                       |
| 710 | Yasunari et al., 2015). This is because of the stronger solar insolation and larger        |                       |
| 711 | albedo reduction due to snow aging, aerosol accumulation within snow, and                  |                       |
| 712 | feedbacks in spring. Dust emissions, and consequent dust transport and deposition, are     |                       |
| 713 | higher in spring than in winter, which may also partly contribute to the larger            |                       |
| 714 | dust-induced SRE in spring than in winter. BC-induced SRE is somewhat larger than          |                       |
|     |                                                                                            |                       |

24

Chenglai 11/12/17 10:31 PM

Chenglai 11/29/17 10:40 PM Deleted: and Chenglai 11/12/17 10:33 PM Deleted: There are more d

Chenglai 11/12/17 10:33 PM Deleted: For different aerosols,

[revised manuscript text omitted]

**Chenglai 11/29/17 11:23 PM**

| 779 | Figure 8 shows surface air temperature, snow water equivalent, and snow                         |                                                           |
|-----|-------------------------------------------------------------------------------------------------|-----------------------------------------------------------|
| 780 | cover fraction changes due to the aerosol SDE in winter and spring, respectively.               |                                                           |
| 781 | Snow water equivalent is defined as the amount of water contained within the                    |                                                           |
| 782 | snowpack, measured kg m -2 which is equivalent to mm after divided by the density of | Chenglai 11/12/17 10:55 PM                                |
| 783 | water (1000 kg $m^{-3}$ ). Snow cover fraction is defined as the fraction of surface area       | Deleted: by
Chenglai 11/12/17 10:57 PM                 |
| 784 | covered by snow. These changes are derived from the difference between the two                  | Deleted: ing                                              |
| 785 | simulations (CTL and NoSDE). The crosses in the figure denote the regions where                 |                                                           |
| 786 | changes are statistically significant at the 0.1 level. Although SRE is largest over the        |                                                           |
| 787 | mountains, surface air temperature change is largest in the regions adjacent to, the            |
Chenglai 11/29/17 11:28 PM                            |
| 788 | mountains, such as over the Eastern Snake River Plain, Northern Utah, and                       | Deleted: around                                           |
| 789 | Central-Southwestern Wyoming, where surface air temperatures are increased by                   |                                                           |
| 790 | around 0.5-2 °C due to the aerosol SDE. The large surface air temperature increase              |                                                           |
| 791 | corresponds well to the significant reductions of snow water equivalent (by 2-50 mm)            |                                                           |
| 792 | and snow cover fraction (by 5-20%) in these regions. This indicates a pronounced                |                                                           |
| 793 | positive feedback between snow albedo, radiation, and surface temperature in the                |                                                           |
| 794 | regions adjacent to the mountains, where snow water equivalent values are relatively            |
Chenglai 11/29/17 11:29 PM                            |
| 795 | lower and snow cover fractions are smaller than those over the mountains. The                   | Deleted: around                                           |
| 796 | positive feedback amplifies the surface warming and snow melting, as was also found             |                                                           |
| 797 | in a previous study using the Weather Research and Forecasting (WRF) model (Qian                |                                                           |
| 798 | et al., 2009). We note, however, both snow water equivalent and snow cover fraction             |                                                           |
| 799 | are larger over the mountains. For example, winter and spring snow water equivalent             |
Chenglai 11/29/17 11:36 PM

| 805 | is mostly above 50 mm on the high mountains (see Figure 8 of Wu et al. (2017)).         |
|-----|-----------------------------------------------------------------------------------------|
| 806 | Local aerosol SDE may also induce substantial impacts on the surface temperature        |
| 807 | and snowpack on the high mountains, but these impacts may be canceled out by the        |
| 808 | increase of snowfall (Figure 9f). As shown in Figure 8, the smaller change of surface   |
| 809 | air temperature over the mountains corresponds well with the increase of snow water     |
| 810 | equivalent and snow cover fraction (especially in the Northern Rockies and Greater      |
| 811 | Yellowstone region),                                                                    |
| 812 | The increase of snowfall in Figure 9 is likely related to the large-scale               |
| 813 | circulation change due to aerosol SDE. Figure 10 shows wintertime tropospheric          |
| 814 | temperature and zonal winds in CTL and NoSDE simulations and their difference. In       |
| 815 | the NoSDE simulation, we have turned off the SDE not only in the Rocky Mountain         |
| 816 | region, but also in other regions of the globe. Due to aerosol SDE, temperature is      |
| 817 | increased in the high-latitudes of northern hemisphere (Figure 10c), which can reduce   |
| 818 | the meridional temperature gradient, thus leading to the weakening polar jet stream     |
| 819 | north of 50 °N (Figure 10f). This suggests a shift to a more meridional wind pattern in |
| 820 | winter, which can enhance the broader meanders and thus the formation of winter         |
| 821 | storms (Wu et al., 2017). Enhanced winter storm activity further reduces surface        |
| 822 | temperature to the north of Rocky Mountain region as well as in the northern part of    |
| 823 | Rocky Mountain region (Figures 8a and 10c). This, together with the increased           |
| 824 | temperature in the southwestern U.S. and southern part of Rocky Mountain region,        |
| 825 | increases meridional temperature gradient and leads to stronger westerly at 30-45°N     |

**Chenglai Wu 11/28/17 9:43 AM $\ensuremath{\textbf{Deleted:}}$ This suggests that snow on the high mountains is less susceptible to the aerosol SDE. Another reason for Chenglai Wu 11/28/17 9:44 AM Deleted: is that Chenglai Wu 11/28/17 9:45 AM Deleted: are increased Chenglai Wu 11/30/17 4:36 PM **Deleted:** due to the increase of snowfall in these regions (Figure 9f), which cancels out the reduction of snow water equivalent resulting from aerosol SDE Chenglai 11/29/17 11:44 PM $\ensuremath{\textbf{Deleted:}}$ due to enhanced water vapor transport from the Pacific Ocean (Figure not shown), which is Chenglai Wu 12/1/17 10:42 AM Deleted: the change of Chenglai Wu 12/1/17 10:42 AM Deleted: experiment Chenglai Wu 11/30/17 4:38 PM Deleted: also**

| 843 | (Figure | 10t). | Stronge | er westerl | y at | t 30-45 | ٩N | tavors | the | water | vapor | trans | port | from | the |
|-----|---------|-------|---------|------------|------|---------|----|--------|-----|-------|-------|-------|------|------|-----|
|     |         |       | -       |            |      |         |    |        |     |       | _     |       |      |      |     |

844 Pacific Ocean. The enhance of winter storm activity and water vapor transport may

- 845 lead to the increase of precipitation (mainly in terms of snowfall in winter). In spring,
- 846 the change in temperature and zonal winds is similar to that in winter, but with a

847 northward shift of the patterns as a result of northward movements of the polar jet

- 848 stream and westerlies, in spring (Figure not shown). Therefore, the change of snowfall
- 849 is likely a result of circulation change induced by SDE from both the Rocky Mountain
- 850 region and remote regions. It is worth isolating the impacts of SDE from the Rocky

851 Mountain region and remote regions (e.g., high-latitudes) in the future. Note that

- 852 increases of snow water equivalent and snow cover fraction in the Northern Rockies
- and Greater Yellowstone region due to aerosol SDE do not pass the significant test at
- 854 0.1 level because of the large interannual variability in these regions.

Table 2 gives the winter and spring surface air temperature changes due to

- 856 LAAs in snow averaged over the five regions. Seasonal mean surface air temperature
- change is around 0.9-1.1 °C in the Eastern Snake River Plain in winter and spring,

858 while this change is around  $\Omega$ -0.2 °C (winter) and around 0.3-0.5 °C (spring) in the

859 mountainous regions (Northern Rockies, Greater Yellowstone region, and Southern

- 860 Rockies). In Table 2, we also show the efficacy of snow albedo forcing, which is
- 861 defined as the ratio of local surface air temperature change to SRE over a specific
- 862 region. The efficacy is mostly around 0.1-0.5 in the three mountainous regions, but it

| Chenglai Wu 12/1/17 10:50 AM |
|------------------------------|
| Deleted: y                   |
| Chenglai Wu 12/1/17 10:39 AM |
| Deleted: local region        |
| Chenglai Wu 12/1/17 10:39 AM |
| Deleted: local               |

Chenglai 11/29/17 11:56 PM Deleted: 0.003-0.17

[revised manuscript text omitted]

Chenglai 11/30/17 12:12 AM                        |
|     |                                                                                                    |                                                                    |

| Deleted: 13 |  |  |
|-------------|--|--|
|             |  |  |
|             |  |  |

Chenglai 11/12/17 11:02 PM Deleted: the Chenglai 11/12/17 11:05 PM Deleted: later month (i.e., Chenglai 11/12/17 11:05 PM Deleted: ) Chenglai 11/30/17 12:11 AM Deleted: With respect to the relative smaller r Chenglai 11/12/17 11:04 PM Deleted: than Chenglai 11/30/17 12:12 AM  $\ensuremath{\textbf{Deleted:}}$  is more significant, which

| 972 | 0.04 (8%), | , 0.17 (23%), | and 0.06 mm day | (16%) in the | three regions, | respectively. |
|-----|------------|---------------|-----------------|--------------|----------------|---------------|
|-----|------------|---------------|-----------------|--------------|----------------|---------------|

973 Note that due to increase of precipitation, the annual mean runoff is increased by 0.12

974 (12%), 0.09 (10%), and 0.01 mm day-1 (2%) in these three regions, respectively.

**975 5. Conclusions**

- 976 In this study, we use VR-CESM to quantify the impacts of LAA\*(BC and dust)
- 977 deposition to the snowpack and hydrologic cycles and to surface air temperatures in
- 978 the Rocky Mountains. Our previous study has shown that VR-CESM reproduces
- 979 reasonably the spatial distributions and seasonal evolution of snowpack in the Rocky
- 980 Mountains (Wu et al., 2017). Here we show that the model simulates similar,
- 981 magnitude of near-surface dust concentrations at most stations in the Rocky Mountain
- 982 region compared to IMPROVE observations. The model tends to underestimate
- 983 near-surface atmospheric BC concentrations mostly by a factor of 1.5-5 in the Rocky
- 984 Mountain region. The underestimation of near-surface BC concentrations may be due
  985 to the absence of local sources in the BC emissions dataset used and too weak.
- 986 transport in the model. Simulated aerosol-in-snow concentrations are closely related
- 987 to the distributions of both snowpack and near-surface atmospheric aerosol
- 988 concentrations. Simulated BC-in-snow concentrations ranges from  $2 t_{0.50} \text{ ng g}^{-1} t_{1.10}$
- 989 the Rocky Mountain region, and they are 35% larger than the observations for the
- 990 average at the 17 sites,
- 991 Due to the deposition of LAAs to snow, surface net shortwave radiation is
- 992 increased. Regional- and seasonal- averaged SRE induced by LAAs in snow is 0.1-0.5

**Chenglai 11/12/17 11:06 PM**

| - | Chenglai Wu 11/30/17 10:09 AM    |     |
|---|----------------------------------|-----|
|   | Deleted: 256              | [4] |
|   | Chenglai 11/30/17 12:45 AM       |     |
|   | Deleted: mostly underestimated,  | [5] |
|   | Chenglai Wu 11/30/17 9:54 AM     |     |
|   | Deleted: er                      |     |
| 1 | Chenglai 11/30/17 12:54 AM       |     |
|   | Deleted:imulated aerosol-in-snow | [6] |

| 1 | Chenglai Wu 11/30/17 10:13 AM |     |
|---|-------------------------------|-----|
|   | Deleted: overall0             | [7] |
| 1 | Chenglai 11/30/17 12:56 AM    |     |
|   | Deleted:                      | ,   |
| 1 | Chenglai 11/12/17 11:15 PM    |     |
|   | Deleted: onsnow cover         | [8] |
| 1 | Chenglai Wu 11/30/17 4:50 PM  |     |
|   | Deleted: ly                   |     |

| 1035 | W m -2 in winter in the three mountainous regions (Northern Rockies, Greater             |                    |
|------|-----------------------------------------------------------------------------------------------------|--------------------|
| 1036 | Yellowstone region, and Southern Rockies) and 0.4-0.6 W $\mathrm{m}^{\text{-2}}$ in the two regions |                    |
| 1037 | around the mountains (Eastern Snake River Plain and Southwestern Wyoming).                          |                    |
| 1038 | Seasonal average SRE is much larger in spring and reaches up to 0.6-1.7 W m -2 in        |                    |
| 1039 | these five regions (Table 2). Dust contributes 21-43% to the total SRE induced by                   |                    |
| 1040 | LAAs in snow in spring, indicating the important role of dust residing in snow. Of the              |                    |
| 1041 | five regions, dust contributes the most (43%) to the total SRE in the Southern Rockies.             |                    |
| 1042 | This is not unexpected as this region is close to dust sources in the Colorado Plateau.             |                    |
| 1043 | As a result of SRE induced by LAAs in snow, surface air temperature                                 |                    |
| 1044 | increases in most of the Rocky Mountain region. The surface air temperature increase                |                    |
| 1045 | is largest over the Eastern Snake River Plain and Southwestern Wyoming, with winter                 |                    |
| 1046 | and spring surface air temperature increased by 0.9-1.1°C. Significant reductions of                |                    |
| 1047 | snow water equivalent (by 2-50 mm) and snow cover fraction (by 5-20%) occur in                      |                    |
| 1048 | these two regions, indicating a strong positive snow-albedo feedback there.                         |                    |
| 1049 | Aerosol SDE accelerates the hydrologic cycle in the mountainous regions. In                         |                    |
| 1050 | April and May, monthly mean runoff is increased by 7%-42% in the three                              |                    |
| 1051 | mountainous regions (Northern Rockies, Greater Yellowstone region, Southern                         |                    |
| 1052 | Rockies). This is because of the accelerated snowmelt resulting from surface warming                |                    |
| 1053 | as well as the increased snowfall resulting from enhanced winter storm activity and                 |                    |
| 1054 | water vapor transport from the Pacific Ocean. This enhancement may be related to                    | Chengl             |
| 1055 | large-scale circulation changes. In the later stage of snowmelt, monthly runoff is                  | Deleted
Chengla |

Chenglai Wu 11/30/17 4:54 PM Deleted: e Chenglai Wu 11/30/17 4:54 PM Deleted: d water vapor transport

| 1058 | reduced by 2-15% in June and 8-23% in July in the three mountainous regions. In             |    |
|------|---------------------------------------------------------------------------------------------|----|
| 1059 | particular, aerosol SDE leads to a reduction of total runoff by about 15% in June and       |    |
| 1060 | July in the Southern Rockies. This highlights the important role of aerosol SDE in          |    |
| 1061 | modulating the hydrologic cycle in these mountainous regions.                               |    |
| 1062 | We note that VR-CESM still underestimates the near-surface BC                               |    |
| 1063 | concentrations, however, overestimates BC-in-snow concentrations by 35% for the             | C  |
| 1064 | average across the 17 observational sites. For dust in snow, the model used in this         | C  |
| 1065 | study only accounts for dust particles smaller than 10 $\mu$ m, while observations made by  | C  |
| 1066 | Reynolds et al. (2016) suggest that most airborne and in-snow dust mass                     | C  |
| 1067 | concentrations are characterized by dust particles with diameters larger than 10 $\mu$ m in | De |
| 1068 | the Utah-Colorado region. Therefore, our simulations may significantly underestimate        |    |
| 1069 | the impacts of dust in snow especially over Southern Rockies. In the Southern               |    |
| 1070 | Rockies, our simulations suggest SRE induced by dust-in-snow can reach up to 2-5 W